# TRADEOFFS IN DATA AUGMENTATION: AN EMPIRICAL STUDY

**Raphael Gontijo-Lopes**[*]
Google Brain
iraphael@google.com

**Sylvia J. Smullin**[*][†]
Blueshift, Alphabet

**Ekin D. Cubuk**
Google Brain
cubuk@google.com

**Ethan Dyer**
Blueshift, Alphabet
edyer@google.com

## ABSTRACT

Though data augmentation has become a standard component of deep neural network training, the underlying mechanism behind the effectiveness of these techniques remains poorly understood. In practice, augmentation policies are often chosen using heuristics of distribution shift or augmentation diversity. Inspired by these, we conduct an empirical study to quantify how data augmentation improves model generalization. We introduce two interpretable and easy-to-compute measures: Affinity and Diversity. We find that augmentation performance is predicted not by either of these alone but by jointly optimizing the two.

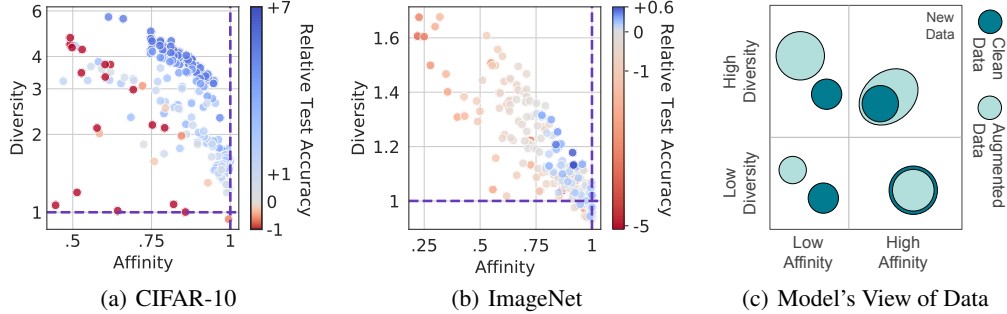

(a) CIFAR-10     (b) ImageNet     (c) Model's View of Data

Figure 1: **Affinity and Diversity parameterize the performance of a model trained with augmentation**. (a, b) Each point represents a different augmentation that yields test accuracy greater than (CIFAR-10: 84.7%, ImageNet: 71.1%). Color shows the final test accuracy relative to the baseline trained without augmentation (CIFAR-10: 89.7%, ImageNet: 76.1%). (c) Representation of how clean data and augmented data are related in the space of these two metrics. Higher diversity is represented by a larger bubble while distributional similarity is depicted through the overlap of bubbles. Test accuracy generally improves to the upper right in this space. Adding real new data to the training set is expected to be in the upper right corner.

## 1 INTRODUCTION

Models that achieve state-of-the-art in image classification often use heavy data augmentation strategies. The best techniques use various transforms applied sequentially and stochastically. Though the effectiveness of this is well-established, the mechanism through which these transformations work is not well-understood.

---

[*]Equal contribution
[†]Currently at Form Energy, Inc, Somerville, MA

Since early uses of data augmentation, it has been assumed that augmentation works because it simulates realistic samples from the true data distribution: "[augmentation strategies are] reasonable since the transformed reference data is now extremely close to the original data. In this way, the amount of training data is effectively increased" (Bellegarda et al., 1992). Because of this, augmentations have often been designed with the heuristic of incurring minimal distribution shift from the training data.

This rationale does not explain why unrealistic distortions such as cutout (DeVries & Taylor, 2017), SpecAugment (Park et al., 2019), and mixup (Zhang et al., 2017) significantly improve generalization performance. Furthermore, methods do not always transfer across datasets—`Cutout`, for example, is useful on CIFAR-10 and not on ImageNet (Lopes et al., 2019). Additionally, many augmentation policies heavily modify images by stochastically applying multiple transforms to a single image. Based on this observation, some have proposed that augmentation strategies are effective because they increase the diversity of images seen by the model.

In this complex landscape, claims about diversity and distributional similarity remain unverified heuristics. Without more precise data augmentation science, finding state-of-the-art strategies requires brute force that can cost thousands of GPU hours (Cubuk et al., 2018; Zhang et al., 2019). This highlights a need to specify and measure the relationship between the original training data and the augmented dataset, as relevant to a given model's performance.

In this paper, we quantify these heuristics. Seeking to understand the mechanisms of augmentation, we focus on single transforms as a foundation. We present an empirical study of 204 different augmentations on CIFAR-10 and 225 on ImageNet, varying both broad transform families and finer transform parameters. To better understand current state of the art augmentation policies, we additionally measure 58 composite augmentations on ImageNet and three state of the art augmentations on CIFAR-10. Our contributions are:

1. We introduce Affinity and Diversity: interpretable, easy-to-compute metrics for parametrizing augmentation performance. Affinity quantifies how much an augmentation shifts the training data distribution. Diversity quantifies the complexity of the augmented data with respect to the model and learning procedure.

2. We find that performance is dependent on *both* metrics. In the Affinity-Diversity plane, the best augmentation strategies jointly optimize the two (see Fig 1).

3. We connect augmentation to other familiar forms of regularization, such as $\ell_2$ and learning rate scheduling, observing common features of the dynamics: performance can be improved and training accelerated by turning off regularization at an appropriate time.

4. We find that performance is only improved when a transform increases the total number of unique training examples. The utility of these new training examples is informed by the augmentation's Affinity and Diversity.

## 2 RELATED WORK

Since early uses of data augmentation in training neural networks, there has been an assumption that effective transforms for data augmentation are those that produce images from an "overlapping but different" distribution (Bengio et al., 2011; Bellegarda et al., 1992). Indeed, elastic distortions as well as distortions in the scale, position, and orientation of training images have been used on MNIST (Ciregan et al., 2012; Sato et al., 2015; Simard et al., 2003; Wan et al., 2013), while horizontal flips, random crops, and random distortions to color channels have been used on CIFAR-10 and ImageNet (Krizhevsky et al., 2012; Zagoruyko & Komodakis, 2016; Zoph et al., 2017). For object detection and image segmentation, one can also use object-centric cropping (Liu et al., 2016) or cut-and-paste new objects (Dwibedi et al., 2017; Fang et al., 2019; Ngiam et al., 2019).

In contrast, researchers have also successfully used less domain-specific transformations, such as Gaussian noise (Ford et al., 2019; Lopes et al., 2019), input dropout (Srivastava et al., 2014), erasing random patches of the training samples (DeVries & Taylor, 2017; Park et al., 2019; Zhong et al., 2017), and adversarial noise (Szegedy et al., 2013). Mixup (Zhang et al., 2017) and Sample Pairing (Inoue, 2018) are two augmentation methods that use convex combinations of training samples.

It is also possible to improve generalization by combining individual transformations. For example, reinforcement learning has been used to choose more optimal combinations of data augmentation transformations (Ratner et al., 2017; Cubuk et al., 2018). Follow-up research has lowered the computation cost of such optimization, by using population based training (Ho et al., 2019), density matching (Lim et al., 2019), adversarial policy-design that evolves throughout training (Zhang et al., 2019), or a reduced search space (Cubuk et al., 2019). Despite producing unrealistic outputs, such combinations of augmentations can be highly effective in different tasks (Berthelot et al., 2019; Tan & Le, 2019; Tan et al., 2019; Xie et al., 2019a;b).

Across these different examples, the role of distribution shift in training remains unclear. Lim et al. (2019); Hataya et al. (2020) have found augmentation policies by minimizing the distance between the distributions of augmented data and clean data. Recent work found that after training with augmented data, fine-tuning on clean training data can be beneficial (He et al., 2019), while Touvron et al. (2019) found it beneficial to fine-tune with a test-set resolution that aligns with the training-set resolution.

The true input-space distribution from which a training dataset is drawn remains elusive. To better understand the effect of distribution shift on performance, many works attempt to estimate it. Often these techniques require training secondary models, such as those based on variational methods (Goodfellow et al., 2014; Kingma & Welling, 2014; Nowozin et al., 2016; Blei et al., 2017). Others have augmented the training set by modelling the data distribution directly (Tran et al., 2017). Recent work has suggested that even unrealistic distribution modelling can be beneficial (Dai et al., 2017).

These methods try to specify the distribution separately from the model they are trying to optimize. As a result, they are insensitive to any interaction between the model and data distribution. Instead, we are interested in a measure of how much the data shifts along directions that are most relevant to the model's performance.

## 3 METHODS

We performed extensive experiments with various augmentations on CIFAR-10 and ImageNet. Experiments on CIFAR-10 used the WRN-28-2 model (Zagoruyko & Komodakis, 2016), trained for 78k steps with cosine learning rate decay. Results are the mean over 10 initializations and reported errors (often too small to show on figures) are the standard error on the mean. Details on the error analysis are in Sec. C.

Experiments on ImageNet used the ResNet-50 model (He et al., 2016), trained for 112.6k steps with a weight decay rate of 1e-4, and a learning rate of 0.2, which is decayed by 10 at epochs 30, 60, and 80.

Images were pre-processed by dividing each pixel value by 255 and normalizing by the dataset statistics. Random crop was also applied on all ImageNet models. These pre-processed data without further augmentation are "clean data" and a model trained on it is the "clean baseline". We followed the same implementation details as Cubuk et al. (2018)[1], including for most augmentation operations. Further implementation details are in Sec. A.

Unless specified otherwise, data augmentation was applied following standard practice: each time an image is drawn, the given augmentation is applied with a given probability. We call this mode *dynamic* augmentation. Due to whatever stochasticity is in the transform itself (such as randomly selecting the location for a crop) or in the policy (such as applying a flip only with 50% probability), the augmented image could be different each time. Thus, most of the tested augmentations increase the number of possible distinct images that can be shown during training.

We also performed select experiments by training with *static* augmentation. In static augmentation, the augmentation policy (one or more transforms) is applied once to the entire clean training set. Static augmentation does not change the number of unique images in the dataset.

### 3.1 AFFINITY: A SIMPLE METRIC FOR DISTRIBUTION SHIFT

Thus far, heuristics of distribution shift have motivated design of augmentation policies. Inspired by this focus, we introduce a simple metric to quantify how augmentation shifts data *with respect to the decision boundary of the clean baseline model*.

---

[1]Available at `bit.ly/2v2FojN`

We start by noting that a trained model is often sensitive to the distribution of the training data. That is, model performance varies greatly between new samples from the true data distribution and samples from a shifted distribution.

Importantly, the model's sensitivity to distribution shift is not purely a function of the input data distribution, since training dynamics and the model's implicit biases affect performance. Because the goal of augmentation is improving model performance, measuring shifts with respect to the distribution captured by the model is more meaningful than measuring shifts in the distribution of the input data alone.

We thus define Affinity to be the ratio between the validation accuracy of a model trained on clean data and tested on an augmented validation set, and the accuracy of the same model tested on clean data. Here, the augmentation is applied to the validation dataset in one pass, as a static augmentation. More formally we define:

**Definition 1.** *Let $D_{train}$ and $D_{val}$ be training and validation datasets drawn IID from the same clean data distribution, and let $D'_{val}$ be derived from $D_{val}$ by applying a stochastic augmentation strategy, a, once to each image in $D_{val}$, $D'_{val} = \{(a(x), y) : \forall (x, y) \in D_{val}\}$. Further let $m$ be a model trained on $D_{train}$ and $\mathcal{A}(m, D)$ denote the model's accuracy when evaluated on dataset $D$. The Affinity, $\mathcal{T}[a; m; D_{val}]$, is given by*

$$\mathcal{T}[a; m; D_{val}] = \mathcal{A}(m, D'_{val})/\mathcal{A}(m, D_{val}) .\tag{1}$$

With this definition, Affinity of one represents no shift and a smaller number suggests that the augmented data is out-of-distribution for the model.

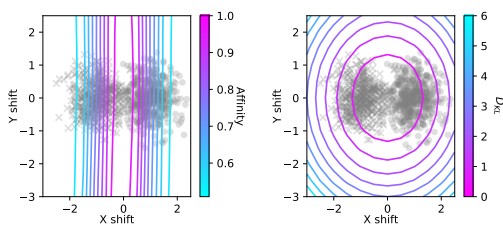

Figure 2: **Affinity is a model-sensitive measure of distribution shift**. Contours indicate lines of equal (left) Affinity, or (right) KL Divergence between the joint distribution of the original data and targets and the shifted data. The two axes indicate the actual shifts that define the augmentation. Affinity captures model-dependent features, such as the decision boundary.

In Fig. 2 we illustrate Affinity with a two-class classification task on a mixture of two Gaussians. Augmentation in this example comprises shift of the means of the Gaussians of the validation data compared to those used for training. Under this shift, we calculate both Affinity and KL divergence of the shifted data with respect to the original data. Affinity changes only when the shift in the data is with respect to the model's decision boundary, whereas the KL divergence changes even when data is shifted in the direction that is irrelevant to the classification task. In this way, Affinity captures what is relevant to a model: shifts that impact predictions.

This same metric has been used as a measure of a model's robustness to image corruptions that do not change images' semantic content (Azulay & Weiss, 2018; Dodge & Karam, 2017; Ford et al., 2019; Hendrycks & Dietterich, 2019; Rosenfeld et al., 2018; Yin et al., 2019). Here we, turn this around and use it to quantify the shift of augmented data compared to clean data.

Affinity has the following advantages as a metric:

1. It is easy to measure. It requires only clean training of the model in question.
2. It is independent of any confounding interaction between the data augmentation and the training process, since augmentation is only used on the validation set and applied statically.
3. It is a measure of distance sensitive to properties of both the data distribution *and* the model.

We gain confidence in this metric by comparing it to other potential model-dependent measures of distribution shift. We consider the mean log likelihood of augmented test images (Grathwohl et al., 2019), and the Watanabe–Akaike information criterion (WAIC) (Watanabe, 2010). These other metrics have high correlation with Affinity. Details can be found in Sec. F.

## 3.2 DIVERSITY: A MEASURE OF AUGMENTATION COMPLEXITY

Inspired by the observation that multi-factor augmentation policies such as `FlipLR+Crop+Cutout` and `RandAugment`(Cubuk et al., 2019) greatly improve performance, we propose another axis on

which to view augmentation policies, which we dub *Diversity*. This measure is intended to quantify the intuition that augmentations prevent models from over-fitting by increasing the number of samples in the training set; the importance of this is shown in Sec. 4.3.

Based on the intuition that more diverse data should be more difficult for a model to fit, we propose a model-based measure. The Diversity metric in this paper is the ratio of the final training loss of a model trained with a given augmentation, relative to the final training loss of the model trained on clean data:

**Definition 2.** *Let $a$ be an augmentation and $D'_{train}$ be the augmented training data resulting from applying the augmentation, $a$, stochastically. Further, let $L_{train}$ be the training loss for a model, $m$, trained on $D'_{train}$. We define the Diversity, $\mathcal{D}[a; m; D_{train}]$, as*

$$\mathcal{D}[a; m; D_{train}] := \mathbb{E}_{D'_{train}}[L_{train}] / \mathbb{E}_{D_{train}}[L_{train}] \ . \tag{2}$$

As with Affinity, this definition of Diversity has the advantage that it can capture model-dependent elements, i.e. it is informed by the class of priors implicit in choosing a model and optimization scheme as well as by the stopping criterion used in training.

One shortcoming of the above measure is the computational cost of evaluation. For each augmentation strategy of interest one must train an independent model and computing the training loss is as computationally costly as determining final test accuracy. For some purposes this cost may not be a problem. For instance, many successful augmentation strategies are usable across a wide range of models. This re-usability helps differ the initial search cost. Here we focus on this metric as a tool for understanding.

Nonetheless, with computational cost in mind, in Sec. E we consider two alternative measures of diversity that are substantially less costly to evaluate. The first of these is again using the training loss but at an earlier point during training. We find that even after ten epochs of training the early time loss correlates well with its value at the end of training.

Another potential diversity measure is the entropy of the transformed data, $\mathcal{D}_{\text{Ent}}$. This is inspired by the intuition that augmentations with more degrees of freedom perform better. For discrete transformations, we consider the conditional entropy of the augmented data.

$$\mathcal{D}_{\text{Ent}} := H(X'|X) = -\mathbb{E}_X \left[ \Sigma_{x'} p(x'|X) \log(p(x'|X)) \right] \ .$$

Here $x \in X$ is a clean training image and $x' \in X'$ is an augmented image. This measure has the property that it can be evaluated without any training or reference to model architecture. However, the appropriate entropy for continuously-varying transforms is less straightforward.

Lastly, in Sec. E we consider a fourth proxy for Diversity, the training time needed for a model to reach a given training accuracy threshold. In Sec. E, we show that these metrics correlate well with each other.

In the remaining sections we describe how the complementary metrics of Diversity and Affinity can be used to characterize and understand augmentation performance.

## 4 RESULTS

### 4.1 AUGMENTATION PERFORMANCE IS DETERMINED BY BOTH AFFINITY AND DIVERSITY

Despite the original inspiration to mimic realistic transformations and minimize distribution shift, many state-of-the-art augmentations yield unrealistic images. This suggests that distribution shift alone does not fully describe or predict augmentation performance.

Figure 3(a) (left) measures Affinity across 204 different augmentations for CIFAR-10 and 223 for ImageNet respectively. We find that for the most important augmentations—those that help performance—Affinity is a poor predictor of accuracy. Furthermore, we find many successful augmentations with low Affinity. For example, `Rotate(fixed, 45deg, 50%)`, `Cutout(16)`, and combinations of `FlipLR`, `Crop(32)`, and `Cutout(16)` all have Affinity$< 0.83$ and test accuracy$>$ 2% above clean baseline on CIFAR-10. Augmentation details are in Sec. B.

As Affinity does not fully characterize the performance of an augmentation, we seek another metric. To assess the importance of an augmentation's complexity, we measure Diversity across the same set

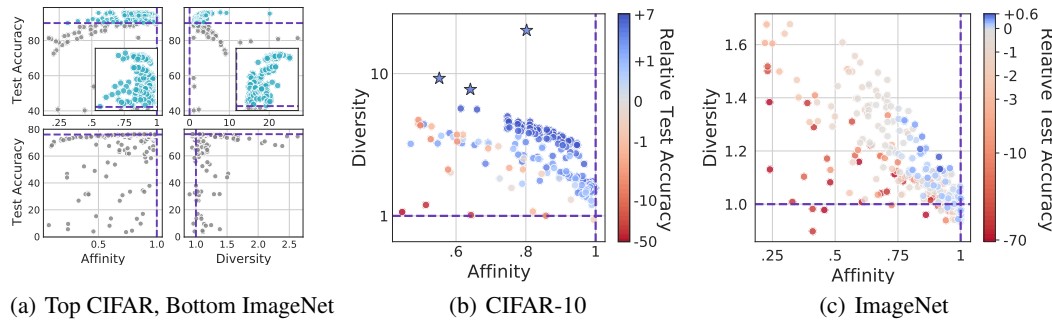

(a) Top CIFAR, Bottom ImageNet     (b) CIFAR-10     (c) ImageNet

Figure 3: **Augmentation performance is determined by both Affinity and Diversity**. (a) Test accuracy plotted against each of Affinity and Diversity for the two datasets, showing that neither metric alone predicts performance. In the CIFAR-10 plots (top), blue highlights (also in inset) are the augmentations that increase test accuracy above the clean baseline. Dashed lines indicate the clean baseline. (b) and (c) show test accuracy relative to clean baseline on the color scale in the plane of Affinity and Diversity. The three star markers in (b) are (left to right) `RandAugment`, `AutoAugment`, and `mixup`. For fixed values of Affinity, test accuracy generally increases with higher values of Diversity. For fixed values of Diversity, test accuracy generally increases with higher values of Affinity. To quantify this tendency we measured what fraction of points satisfy Inequality 3. For CIFAR-10 99.1% of point pairs satisfy the inequality, while for ImageNet 97.5% satisfy it.

of augmentations. We find that Diversity is complementary in explaining how augmentations can increase test performance. As shown in Fig. 3(b) and (c), Affinity and Diversity together provide a much clearer parameterization of an augmentation policy's benefit to performance. For a fixed level of Diversity, augmentations with higher Affinity are consistently better. Similarly, for a fixed Affinity, it is generally better to have higher Diversity. To make this more quantitative we consider pairs of augmentation strategies $(a, a')$ and measure the fraction of pairs satisfying

$$\{(a, a') : \mathcal{A}(a) > \mathcal{A}(a') \text{ and } \mathcal{T}(a) > \mathcal{T}(a') \text{ or } \mathcal{D}(a) > \mathcal{D}(a')\} \tag{3}$$

A simple case study is presented in Fig. 4. The probability of the transform `Rotate(fixed, 60deg)` is varied. The accuracy and Affinity are not monotonically related, with the peak accuracy falling at an intermediate value of Affinity. Similarly, accuracy is correlated with Diversity for low probability transformations, but does not track for higher probabilities. The optimal probability for `Rotate(fixed, 60deg)` lies at an intermediate value of Affinity and Diversity.

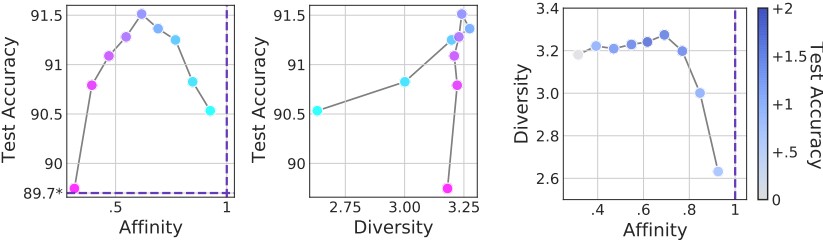

Figure 4: **Test accuracy varies differently than either Affinity or Diversity**. Here, the probability of `Rotate(fixed, 60deg)` on CIFAR-10 is varied from 10% (cyan) to 90% (pink). Left: as probability increases, Affinity decreases linearly while the accuracy changes non-monotonically. Center: accuracy and Diversity vary differently from each other as probability is changed. Right: test accuracy is maximized at intermediate values.

To situate the tested augmentations—mostly single transforms—within the context of the state-of-the-art, we tested three high-performance augmentations from literature: `mixup` (Zhang et al., 2017), `AutoAugment` (Cubuk et al., 2018), and `RandAugment` (Cubuk et al., 2019). These are highlighted with a star marker in Fig. 3(b).

More than either of the metrics alone, Affinity and Diversity together provide a useful parameterization of an augmentation's performance. We now turn to investigating the utility of this tool for explaining other observed phenomena of data augmentations.

## 4.2 TURNING AUGMENTATIONS OFF MAY ADJUST AFFINITY, DIVERSITY, AND PERFORMANCE

The term "regularizer" is ill-defined in the literature, often referring to any technique used to reduce generalization error without necessarily reducing training error (Goodfellow et al., 2016). With this definition, it is widely acknowledged that commonly-used augmentations act as regularizers (Hernández-García & König, 2018a;b; Zhang et al., 2016; Dao et al., 2019). Though this is a broad definition, we notice another commonality across seemingly different kinds of regularizers: various regularization techniques yield boosts in performance (or at least no degradation) if the regularization is *turned off* at the right time during training. For instance:

1. Decaying a large learning rate on an appropriate schedule can be better than maintaining a large learning rate throughout training (Zagoruyko & Komodakis, 2016).

2. Turning off $\ell_2$ at the right time does not hurt performance (Golatkar et al., 2019).

3. Relaxing model constraints mid-training can boost performance (d'Ascoli et al., 2019).

4. Fine-tuning on clean data can improve final accuracy (He et al., 2019).

To further study augmentation as a regularizer, we compare the constant augmentation case (with the same augmentation throughout) to the case where the augmentation is turned off partway through training and training is completed with clean data. For each transform, we test over a range of switch-off points and select the one that yields the best final validation or test accuracy on CIFAR-10 and ImageNet respectively. The Switch-off Lift is the resulting increase in final test accuracy, compared to training with augmented data the entire time.

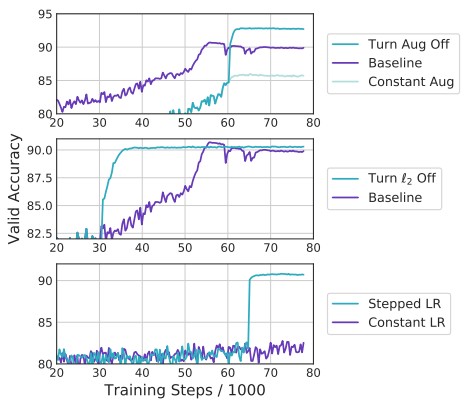

(a) Slingshot effect on CIFAR-10

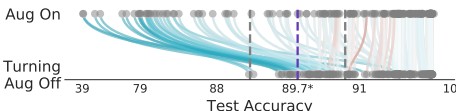

(b) Switch-off Lift on CIFAR-10

Figure 5: **(a) Switching off regularizers yields a performance boost**: Three examples of how turning off a regularizer increases the validation accuracy. This slingshot effect can speed up training and improve the best validation accuracy. Top: training with no augmentation (clean baseline), compared to constant augmentation, and augmentation that is turned off at 55k steps. Here, the augmentation is `Rotate(fixed, 20deg,100%)`. Middle: Baseline with constant $\ell_2$. This is compared to turning off $\ell_2$ regularization part way through training. Bottom: Constant learning rate of 0.1 compared to training where the learning rate is decayed in one step by a factor of 10. **(b) Bad augmentations can become helpful if switched off**: Colored lines connect the test accuracy with augmentation applied throughout training (top) to the test accuracy with switching mid-training. Color indicates the amount of Switch-off Lift; blue is positive and orange is negative. The dashed purple line indicates accuracy of clean model. Crossing it from the left means the switched augmentation outperforms the clean model. Dashed grey lines indicate the threshold for SymLog scaling. The standard error on the mean over the ten independent runs are so small as to not be visible.

For some poor-performing augmentations, this gain can actually bring the test accuracy above the baseline, as shown in Fig. 5(b). We additionally observe (Fig. 5(a)) that this test accuracy improvement can happen quite rapidly for both augmentations and for the other regularizers tested. This suggests an opportunity to accelerate training without hurting performance by appropriately switching off regularization. We call this a *slingshot* effect.

Interestingly, we find the best time for turning off an augmentation is not always close to the end of training, contrary to what is shown in He et al. (2019). For example, without switching, `FlipUD(100%)` (vertically flipping every image) decreases test accuracy by almost 50% compared to clean baseline. When the augmentation is used for only the first third of training, final test accuracy is above the baseline.

He et al. (2019) hypothesized that the gain from turning augmentation off is due to recovery from a distribution shift. Indeed, for many detrimental transformations, the test accuracy gained by turning off the augmentation merely recovers the clean baseline performance. A few of the tested augmentations, such as `FlipLR(100%)`, are fully deterministic. Thus, each time an image is drawn in training, it is augmented the same way. When such an augmentation is turned off partway through training, the model then sees images—the clean ones—that are now new. Indeed, when `FlipLR(100%)` is switched off at the right time, its final test accuracy exceeds that of `FlipLR(50%)` without switching. In this way, switching augmentation off may adjust for not only low Affinity but also low Diversity.

### 4.3    INCREASED EFFECTIVE TRAINING SET SIZE IS CRUCIAL FOR DATA AUGMENTATION

Most augmentations we tested and those used in practice have inherent stochasticity and thus may alter a given training image differently each time the image is drawn. In the typical *dynamic* training mode, these augmentations increase the number of unique inputs seen across training epochs.

To further study how augmentations act as regularizers, we seek to discriminate this increase in effective dataset size from other effects. We train models with *static* augmentation, as described in Sec. 3. This altered training set is used without further modification during training so that the number of unique training inputs is the same between the augmented and the clean training settings.

For almost all tested augmentations, using static augmentation yields lower test accuracy than the clean baseline. Where static augmentation shows a gain (versions of `crop`), the difference is less than the standard error on the mean. As in the dynamic case, poorer performance in the static case is for transforms that have lower Affinity and lower Diversity.

Static augmentations also always perform worse than their non-deterministic, dynamic counterparts, as shown in Fig. 6. This may be because the Diversity of a static augmentation is always less than the dynamic case (see also Sec. E). The decrease in Diversity in the static case suggests a connection between Diversity and the number of training examples.

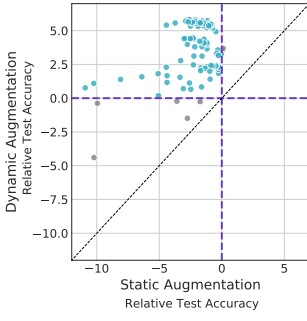 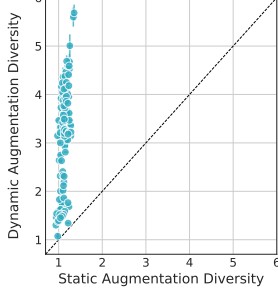

Figure 6: **Static augmentations decrease diversity and performance**. CIFAR-10, static augmentation performance is less than the clean baseline, $(0, 0)$, and less than the dynamic augmentation case (left). Diversity in the static case is less than in the dynamic case (right). Augmentations with no stochasticity are excluded because they are trivially equal on the two axes. Diagonal line indicates where static and dynamic cases would be equal.

Together, these results point to the following conclusion: *Increased effective training set size is crucial to the performance benefit of data augmentation. An augmentation's Affinity and Diversity inform how useful the additional training examples are.*

## 5    DISCUSSION

In this work, we conducted an empirical study in an attempt to characterize the essential factors leading to successful augmentation policies. This builds a foundation for using these metrics to quantify and design more complex and powerful combinations of augmentations. We further hope that Affinity and Diversity may serve as useful metrics of distribution shift more generally.

We believe that our analysis sheds light on the mechanisms of vision augmentations, such as mixup, Cutout, and AutoAugment. Though earlier work has often explicitly focused on just one of these metrics, chosen priors have implicitly ensured reasonable values for both. One way to achieve Diversity is to use combinations of many single augmentations, as in AutoAugment (Cubuk et al., 2018). Because transforms and hyperparameters in Cubuk et al. (2018) were chosen by optimizing performance on proxy tasks, the optimal policies include high and low Affinity transforms. Fast AutoAugment (Lim et al., 2019), CTAugment (Berthelot et al., 2019; Sohn et al., 2020), and

differentiable RandAugment (Cubuk et al., 2019) all aim to increase Affinity by what Lim et al. (2019) called "density-matching". However these methods use the search space of AutoAugment and thus inherit its Diversity.

On the other hand, Adversarial AutoAugment (Zhang et al., 2019) focused on increasing Diversity by optimizing policies to increase the training loss. While this method did not explicitly aim to increase Affinity, it also used transforms and hyperparameters from the AutoAugment search space which implicitly have higher Affinity than random transformations one can apply to an image. For example, rotations are restricted to -30 to 30 degrees in AutoAugment, which have higher Affinity than a random rotation. Without such a prior, the goal of maximizing training loss with no other constraints would lead to data augmentation policies that erase all the information from the images.

Our work is the first one to propose and study these two metrics generally, which helps put previous work in context. Furthermore, our work suggests that further improvements can be expected if one tries to increase both of the metrics, instead of just one of them as was the case in previous work. Given our work, an obvious strategy to find better policies would be to combine the losses of Adversarial AutoAugment and Fast AutoAugment, which we plan to do in future work. Finally, and knowing our insights from the vision domain, we speculate that a similar mechanism might be at play in other domains, such as speech recognition, and underlie why strategies such as SpecAugment, which appear to be relatively out of distribution, nonetheless work – we predict that they drastically increase diversity.

## 6 CONCLUSION

We attempted to quantify common intuition that more in-distribution and more diverse augmentation policies perform well. To this end, we introduced two easy-to-compute metrics, Affinity and Diversity, intended to measure to what extent a given augmentation is in-distribution and how complex the augmentation is to learn. Because they are model-dependent, these metrics capture the data shifts that affect model performance. We hope our findings provide a foundation for continued scientific study of data augmentation and more general distribution shifts.

## ACKNOWLEDGEMENTS

The authors would like to thank Alex Alemi, Justin Gilmer, Guy Gur-Ari, Albin Jones, Behnam Neyshabur, Zan Armstrong, and Ben Poole for thoughtful discussions on this work.

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

# SUPPLEMENTARY MATERIAL

## A    TRAINING METHODS

Cifar10 models were trained using code based on AutoAugment code[2] using the following choices:

1. Learning rate was decayed following a cosine decay schedule, starting with a value of 0.1
2. 78050 training steps were used, with data shuffled after every epoch.
3. As implemented in the AutoAugment code, the WRN-28-2 model was used with stochastic gradient descent and momentum. The optimizer used cross entropy loss with $\ell_2$ weight decay of 0.0005.
4. Before selecting the validation set, the full training set was shuffled and balanced such that the subset selected for training was balanced across classes.
5. Validation set was the last 5000 samples of the shuffled CIFAR-10 training data.
6. Models were trained using Python 2.7 and TensorFlow 1.13 .

A training time of 78k steps was chosen because it showed reasonable convergence with the standard data augmentation of `FlipLR`, `Crop`, and `Cutout` In the clean baseline case, test accuracy actually reached its peak much earlier than 78k steps.

With CIFAR-10, experiments were also performed for training dataset sizes of 1024, 4096, and 16384. At smaller dataset sizes, the impact of augmentation and the Switch-off Lift tended to be larger. These results are not shown in this paper.

ImageNet models were ResNet-50 trained using the Cloud TPU codebase[3]. Models were trained for 112.6k steps with a weight decay rate of 1e-4, and a learning rate of 0.2, which was decayed by 10 at epochs 30, 60, and 80. Batch size was set to be 1024.

## B    DETAILS OF AUGMENTATION

### B.1    CIFAR-10

On CIFAR-10, both color and affine transforms were tested, as given in the full results (see Sec. G). Most augmentations were as defined in Cubuk et al. (2018) and additional conventions for augmentations as labeled in Fig. 7 are defined here. For `Rotate`, *fixed* means each augmented image was rotated by exactly the stated amount, with a randomly-chosen direction. *Variable* means an augmented image was rotated a random amount up to the given value in a randomly-chosen direction. `Shear` is defined similarly. `Rotate(square)` means that an image was rotated by an amount chosen randomly from [0°, 90°, 180°, 270°].

`Crop` included a padding before the random-location crop so that the final image remained $32 \times 32$ in size. The magnitude given for `Crop` is the number of pixels that were added in each dimension. The magnitude given in the label for `Cutout` is the size, in pixels, of each dimension of the square cutout.

`PatchGaussian` was defined as in Lopes et al. (2019), with the patch specified to be contained entirely within the image domain. In Fig. 7, it is labeled by two hyperparameters: the size of the square patch (in pixels) that was applied and $\sigma_{max}$, which is the maximum standard deviation of the noise that could be selected for any given patch. Here, "fixed" means the patch size was always the same.

Since `FlipLR`, `Crop`, and `Cutout` are part of standard pipelines for CIFAR-10 (Hernández-García & König, 2018a; Goodfellow et al., 2013; Springenberg et al., 2014), we tested combinations of the three augmentations (varying probabilities of each) as well as these three augmentations plus an single additional augmentation. As in standard processing of CIFAR-10 images, the first augmentation applied was anything that is not one of `FlipLR`, `Crop`, or `Cutout`. After that, augmentations were applied in the order `Crop`, then `FlipLR`, then `Cutout`.

---

[2]available at `github.com/tensorflow/models/tree/master/research/autoaugment`
[3]available at `github.com/tensorflow/tpu/tree/master/models/official/resnet`

Finally, we tested the CIFAR-10 AutoAugment policy (Cubuk et al., 2018), RandAugment (Cubuk et al., 2019), and mixup (Zhang et al., 2017). The hyperparameters for these augmentations followed the guidelines described in the respective papers.

These augmentations are labeled in Fig. 7.

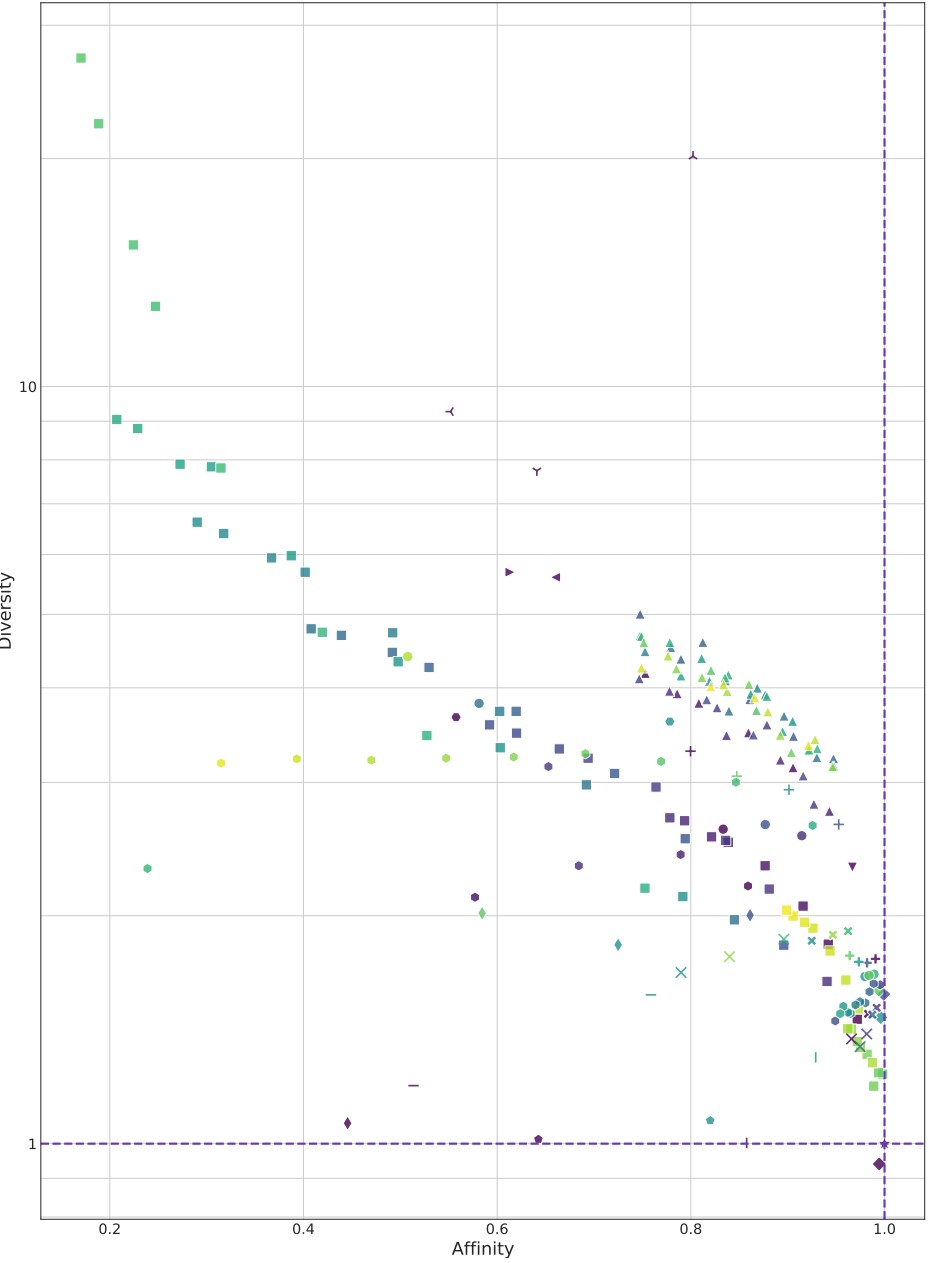

Figure 7: CIFAR-10: Labeled map of tested augmentations on the plane of Affinity and Diversity. Color distinguishes different hyperparameters for a given transform. Legend is below.

Clean
Autoaugment
Mixup
Randaug
PatchGaussian(fixed,12, 0.1, 100%)
PatchGaussian(fixed,12, 0.2, 100%)
PatchGaussian(fixed,12, 0.3, 100%)
PatchGaussian(fixed,12, 0.5, 100%)
PatchGaussian(fixed,12, 0.8, 100%)
PatchGaussian(fixed,12, 1.0, 100%)
PatchGaussian(fixed,12, 1.5, 100%)
PatchGaussian(fixed,12, 2.0, 100%)
PatchGaussian(fixed,16, 0.1, 100%)
PatchGaussian(fixed,16, 0.2, 100%)
PatchGaussian(fixed,16, 0.3, 100%)
PatchGaussian(fixed,16, 0.5, 100%)
PatchGaussian(fixed,16, 0.8, 100%)
PatchGaussian(fixed,16, 1.0, 100%)
PatchGaussian(fixed,16, 1.5, 100%)
PatchGaussian(fixed,16, 2.0, 100%)
PatchGaussian(fixed,20, 0.1, 100%)
PatchGaussian(fixed,20, 0.2, 100%)
PatchGaussian(fixed,20, 0.3, 100%)
PatchGaussian(fixed,20, 0.5, 100%)
PatchGaussian(fixed,20, 0.8, 100%)
PatchGaussian(fixed,20, 1.0, 100%)
PatchGaussian(fixed,20, 1.5, 100%)
PatchGaussian(fixed,20, 2.0, 100%)
PatchGaussian(fixed,24, 0.1, 100%)
PatchGaussian(fixed,24, 0.2, 100%)
PatchGaussian(fixed,24, 0.3, 100%)
PatchGaussian(fixed,24, 0.5, 100%)
PatchGaussian(fixed,24, 0.8, 100%)
PatchGaussian(fixed,24, 1.0, 100%)
PatchGaussian(fixed,24, 1.5, 100%)
PatchGaussian(fixed,24, 2.0, 100%)
PatchGaussian(fixed,28, 0.1, 100%)
PatchGaussian(fixed,28, 0.2, 100%)
PatchGaussian(fixed,28, 0.3, 100%)
PatchGaussian(fixed,28, 0.5, 100%)
PatchGaussian(fixed,28, 0.8, 100%)
PatchGaussian(fixed,28, 1.0, 100%)
PatchGaussian(fixed,28, 1.5, 100%)
PatchGaussian(fixed,28, 2.0, 100%)
PatchGaussian(fixed,32, 0.1, 100%)
PatchGaussian(fixed,32, 0.2, 100%)
PatchGaussian(fixed,32, 0.3, 100%)
PatchGaussian(fixed,32, 0.5, 100%)
PatchGaussian(fixed,32, 0.8, 100%)
PatchGaussian(fixed,32, 1.0, 100%)
PatchGaussian(fixed,32, 1.5, 100%)
PatchGaussian(fixed,32, 2.0, 100%)
PatchGaussian(fixed,4, 0.1, 100%)
PatchGaussian(fixed,4, 0.2, 100%)
PatchGaussian(fixed,4, 0.3, 100%)
PatchGaussian(fixed,4, 0.5, 100%)
PatchGaussian(fixed,4, 0.8, 100%)
PatchGaussian(fixed,4, 1.0, 100%)
PatchGaussian(fixed,4, 1.5, 100%)
PatchGaussian(fixed,4, 2.0, 100%)
PatchGaussian(fixed,8, 0.1, 100%)
PatchGaussian(fixed,8, 0.2, 100%)
PatchGaussian(fixed,8, 0.3, 100%)
PatchGaussian(fixed,8, 0.5, 100%)
PatchGaussian(fixed,8, 0.8, 100%)
PatchGaussian(fixed,8, 1.0, 100%)
PatchGaussian(fixed,8, 1.5, 100%)
PatchGaussian(fixed,8, 2.0, 100%)

Invert(100%)
Invert(50%)
ShearX(variable, 0.1, 100%)
ShearX(variable, 0.1, 50%)
ShearX(variable, 0.1, 75%)
ShearX(variable, 0.3, 100%)
ShearX(variable, 0.3, 50%)
ShearX(variable, 0.3, 75%)
ShearX(fixed ,0.1, 100%)
ShearX(fixed, 0.1, 50%)
ShearX(fixed, 0.1, 75%)
ShearX(fixed, 0.3, 100%)
ShearX(fixed, 0.3, 50%)
ShearX(fixed, 0.3, 75%)
Rotate(variable, 20deg, 100%)
Rotate(variable, 20deg, 50%)
Rotate(variable, 20deg, 75%)
Rotate(variable, 45, 100%)
Rotate(variable, 5deg, 100%)
Rotate(variable, 5deg, 50%)
Rotate(variable, 5deg, 75%)
Rotate(variable, 60deg, 100%)
Rotate(fixed, 15deg, 50%)
Rotate(fixed, 20deg, 100%)
Rotate(fixed, 20deg, 50%)
Rotate(fixed, 20deg, 75%)
Rotate(fixed, 45deg, 50%)
Rotate(fixed, 5deg, 10%)
Rotate(fixed, 5deg, 100%)
Rotate(fixed, 5deg, 20%)
Rotate(fixed, 5deg, 30%)
Rotate(fixed, 5deg, 40%)
Rotate(fixed, 5deg, 50%)
Rotate(fixed, 5deg, 60%)
Rotate(fixed, 5deg, 70%)
Rotate(fixed, 5deg, 75%)
Rotate(fixed, 5deg, 80%)
Rotate(fixed, 5deg, 90%)
Rotate(fixed, 60deg, 10%)
Rotate(fixed, 60deg, 100%)
Rotate(fixed, 60deg, 20%)
Rotate(fixed, 60deg, 30%)
Rotate(fixed, 60deg, 40%)
Rotate(fixed, 60deg, 50%)
Rotate(fixed, 60deg, 60%)
Rotate(fixed, 60deg, 70%)
Rotate(fixed, 60deg, 80%)
Rotate(fixed, 60deg, 90%)
Rotate(square, 100%)
Rotate(square, 50%)
Blur(100%)
Blur(50%)
FlipLR(100%)
FlipLR(25%)
FlipLR(50%)
FlipLR(75%)
FlipUD(100%)
FlipUD(25%)
FlipUD(50%)
FlipUD(75%)
Crop(4, 25%)
Crop(4, 50%)
Crop(4, 75%)
Crop(4,100%)
Cutout(16, 100%)
Cutout(16, 25%)
Cutout(16, 50%)
Cutout(16, 75%)

Equalize(100%)
Equalize(50%)
FlipLR(50%) + Crop(4,100%)
FlipLR(100%) + Crop(4,100%) + Cutout(16,100%)
FlipLR(100%) + Crop(4,100%) + Cutout(16,25%)
FlipLR(100%) + Crop(4,100%) + Cutout(16,50%)
FlipLR(100%) + Crop(4,100%) + Cutout(16,75%)
FlipLR(100%) + Crop(4,25%) + Cutout(16,100%)
FlipLR(100%) + Crop(4,25%) + Cutout(16,25%)
FlipLR(100%) + Crop(4,25%) + Cutout(16,50%)
FlipLR(100%) + Crop(4,25%) + Cutout(16,75%)
FlipLR(100%) + Crop(4,50%) + Cutout(16,100%)
FlipLR(100%) + Crop(4,50%) + Cutout(16,25%)
FlipLR(100%) + Crop(4,50%) + Cutout(16,50%)
FlipLR(100%) + Crop(4,50%) + Cutout(16,75%)
FlipLR(100%) + Crop(4,75%) + Cutout(16,100%)
FlipLR(100%) + Crop(4,75%) + Cutout(16,25%)
FlipLR(100%) + Crop(4,75%) + Cutout(16,50%)
FlipLR(100%) + Crop(4,75%) + Cutout(16,75%)
FlipLR(25%) + Crop(4,100%) + Cutout(16,100%)
FlipLR(25%) + Crop(4,100%) + Cutout(16,25%)
FlipLR(25%) + Crop(4,100%) + Cutout(16,50%)
FlipLR(25%) + Crop(4,100%) + Cutout(16,75%)
FlipLR(25%) + Crop(4,25%) + Cutout(16,100%)
FlipLR(25%) + Crop(4,25%) + Cutout(16,25%)
FlipLR(25%) + Crop(4,25%) + Cutout(16,50%)
FlipLR(25%) + Crop(4,25%) + Cutout(16,75%)
FlipLR(25%) + Crop(4,50%) + Cutout(16,100%)
FlipLR(25%) + Crop(4,50%) + Cutout(16,25%)
FlipLR(25%) + Crop(4,50%) + Cutout(16,50%)
FlipLR(25%) + Crop(4,50%) + Cutout(16,75%)
FlipLR(25%) + Crop(4,75%) + Cutout(16,100%)
FlipLR(25%) + Crop(4,75%) + Cutout(16,25%)
FlipLR(25%) + Crop(4,75%) + Cutout(16,50%)
FlipLR(25%) + Crop(4,75%) + Cutout(16,75%)
FlipLR(50%) + Crop(4,100%) + Cutout(16,100%)
FlipLR(50%) + Crop(4,100%) + Cutout(16,25%)
FlipLR(50%) + Crop(4,100%) + Cutout(16,50%)
FlipLR(50%) + Crop(4,100%) + Cutout(16,75%)
FlipLR(50%) + Crop(4,25%) + Cutout(16,100%)
FlipLR(50%) + Crop(4,25%) + Cutout(16,25%)
FlipLR(50%) + Crop(4,25%) + Cutout(16,50%)
FlipLR(50%) + Crop(4,25%) + Cutout(16,75%)
FlipLR(50%) + Crop(4,50%) + Cutout(16,100%)
FlipLR(50%) + Crop(4,50%) + Cutout(16,25%)
FlipLR(50%) + Crop(4,50%) + Cutout(16,50%)
FlipLR(50%) + Crop(4,50%) + Cutout(16,75%)
FlipLR(50%) + Crop(4,75%) + Cutout(16,100%)
FlipLR(50%) + Crop(4,75%) + Cutout(16,25%)
FlipLR(50%) + Crop(4,75%) + Cutout(16,50%)
FlipLR(50%) + Crop(4,75%) + Cutout(16,75%)
FlipLR(75%) + Crop(4,100%) + Cutout(16,100%)
FlipLR(75%) + Crop(4,100%) + Cutout(16,25%)
FlipLR(75%) + Crop(4,100%) + Cutout(16,50%)
FlipLR(75%) + Crop(4,100%) + Cutout(16,75%)
FlipLR(75%) + Crop(4,25%) + Cutout(16,100%)
FlipLR(75%) + Crop(4,25%) + Cutout(16,25%)
FlipLR(75%) + Crop(4,25%) + Cutout(16,50%)
FlipLR(75%) + Crop(4,25%) + Cutout(16,75%)
FlipLR(75%) + Crop(4,50%) + Cutout(16,100%)
FlipLR(75%) + Crop(4,50%) + Cutout(16,25%)
FlipLR(75%) + Crop(4,50%) + Cutout(16,50%)
FlipLR(75%) + Crop(4,50%) + Cutout(16,75%)
FlipLR(75%) + Crop(4,75%) + Cutout(16,100%)
FlipLR(75%) + Crop(4,75%) + Cutout(16,25%)
FlipLR(75%) + Crop(4,75%) + Cutout(16,50%)
FlipLR(75%) + Crop(4,75%) + Cutout(16,75%)
FlipLR(50%) + Crop(4, 100%) + Cutout(16, 100%) + Equalize(50%)
FlipLR(50%) + Crop(4, 100%) + Cutout(16, 100%) + Rotate(fixed, 15deg, 50%

## B.2 IMAGENET

On ImageNet, we experimented with `PatchGaussian`, `Cutout`, operations from the PIL imaging library[4], and techniques from the AutoAugment code, as described above for CIFAR-10. In addition to `PatchGaussian(fixed)`, we also tested `PatchGaussian(variable)`, where the patch size was uniformly sampled up to a maximum size. The implementation here did not constrain the patch to be entirely contained within the image. Additionally, we experimented with `SolarizeAdd`. `SolarizeAdd` is similar to `Solarize` from the PIL library, but has an additional hyperparameter which determines how much value was added to each pixel that is below the threshold. Finally, we also experimented with `Full Gaussian` and `Random Erasing` on ImageNet. `Full Gaussian` adds Gaussian noise to the whole image. `Random Erasing` is similar to `Cutout`, but randomly

---

[4]https://pillow.readthedocs.io/en/5.1.x/

samples the values of the pixels in the patch (Zhong et al., 2017) (whereas `Cutout` sets them to a constant, gray pixel).

These augmentations are labeled in Fig. 8.

We also experimented with combinations of transformations, using augmentations used in the AutoAugment policy. These are labeled `RandomMultiAug(x, y, z)`, where x is the number of augmentations per subpolicy, y is the number of subpolicies, and z is 0, or 1, indicating two randomized policies ran.

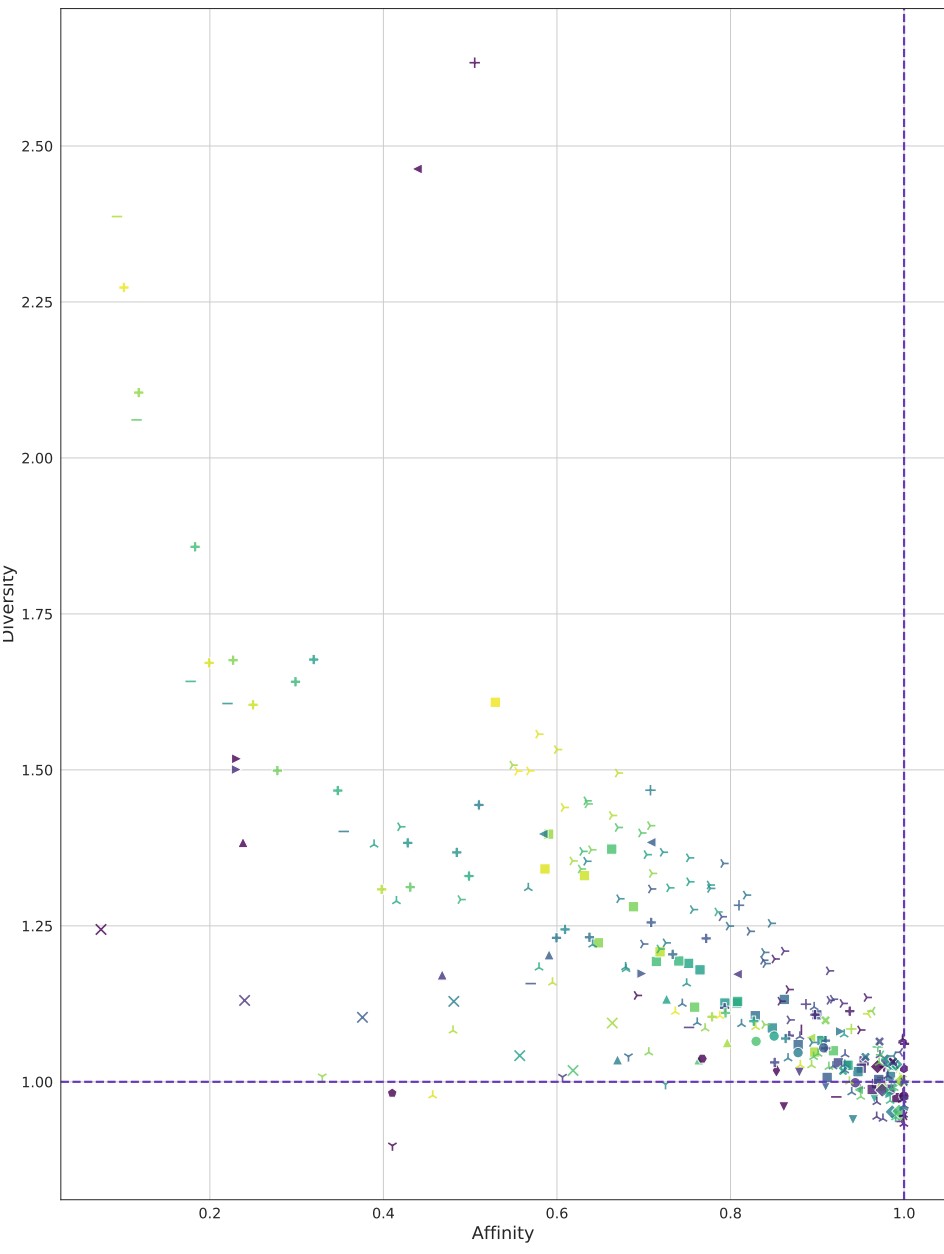

Figure 8: ImageNet: Labeled map of tested augmentations on the plane of Affinity and Diversity. Color distinguishes different hyperparameters for a given transform. Legend is below.

AutoContrast(100%)
Brightness(0.1, 100%)
Brightness(0.2, 100%)
Brightness(0.3, 100%)
Brightness(0.4, 100%)
Brightness(0.5, 100%)
Brightness(0.6, 100%)
Brightness(0.7, 100%)
Clean
Color(0.1, 100%)
Color(0.2, 100%)
Color(0.3, 100%)
Color(0.4, 100%)
Color(0.5, 100%)
Color(0.6, 100%)
Color(0.7, 100%)
Contrast(0.1, 100%)
Contrast(0.2, 100%)
Contrast(0.3, 100%)
Contrast(0.4, 100%)
Contrast(0.5, 100%)
Contrast(0.6, 100%)
Contrast(0.7, 100%)
Cutout(variable, 448, 100%)
Cutout(fixed, 120, 100%)
Cutout(fixed, 150, 100%)
Cutout(fixed, 180, 100%)
Cutout(fixed, 30, 100%)
Cutout(fixed, 60, 100%)
Cutout(fixed, 90, 100%)
Equalize(100%)
FlipUD(100%)
FullGaussian(0.1, 100%)
FullGaussian(0.2, 100%)
FullGaussian(0.3, 100%)
FullGaussian(0.5, 100%)
FullGaussian(0.8, 100%)
FullGaussian(1.0, 100%)
FullGaussian(1.5, 100%)
FullGaussian(2.0, 100%)
Rotate(square, 100%)
Invert(100%)
Patch Gaussian(variable, 100, 0.2, 100%)
Patch Gaussian(variable, 100, 0.5, 100%)
Patch Gaussian(variable, 100, 0.8, 100%)
Patch Gaussian(variable, 100, 1.0, 100%)
Patch Gaussian(variable, 100, 2.0, 100%)
Patch Gaussian(variable, 150, 0.2, 100%)
Patch Gaussian(variable, 150, 0.5, 100%)
Patch Gaussian(variable, 150, 0.8, 100%)
Patch Gaussian(variable, 150, 1.0, 100%)
Patch Gaussian(variable, 150, 2.0, 100%)
Patch Gaussian(variable, 200, 0.2, 100%)
Patch Gaussian(variable, 200, 0.5, 100%)
Patch Gaussian(variable, 200, 0.8, 100%)
Patch Gaussian(variable, 200, 1.0, 100%)
Patch Gaussian(variable, 200, 2.0, 100%)
Patch Gaussian(variable, 250, 0.1, 100%)
Patch Gaussian(variable, 250, 0.2, 100%)
Patch Gaussian(variable, 250, 0.3, 100%)
Patch Gaussian(variable, 250, 0.5, 100%)
Patch Gaussian(variable, 250, 0.8, 100%)
Patch Gaussian(variable, 250, 1.0, 100%)
Patch Gaussian(variable, 250, 1.5, 100%)
Patch Gaussian(variable, 250, 2.0, 100%)
Patch Gaussian(variable, 300, 0.2, 100%)
Patch Gaussian(variable, 300, 0.5, 100%)
Patch Gaussian(variable, 300, 0.8, 100%)
Patch Gaussian(variable, 300, 1.0, 100%)
Patch Gaussian(variable, 300, 2.0, 100%)
Patch Gaussian(variable, 350, 0.2, 100%)
Patch Gaussian(variable, 350, 0.5, 100%)
Patch Gaussian(variable, 350, 0.8, 100%)
Patch Gaussian(variable, 350, 1.0, 100%)
Patch Gaussian(variable, 350, 2.0, 100%)
Patch Gaussian(variable, 400, 0.2, 100%)
Patch Gaussian(variable, 400, 0.5, 100%)
Patch Gaussian(variable, 400, 0.8, 100%)
Patch Gaussian(variable, 400, 1.0, 100%)
Patch Gaussian(variable, 400, 2.0, 100%)
Patch Gaussian(fixed, 100, 0.0, 100%)
Patch Gaussian(fixed, 100, 0.2, 100%)
Patch Gaussian(fixed, 100, 0.5, 100%)
Patch Gaussian(fixed, 100, 0.8, 100%)
Patch Gaussian(fixed, 100, 1.0, 100%)
Patch Gaussian(fixed, 100, 2.0, 100%)
Patch Gaussian(fixed, 150, 0.2, 100%)
Patch Gaussian(fixed, 150, 0.5, 100%)
Patch Gaussian(fixed, 150, 0.8, 100%)
Patch Gaussian(fixed, 150, 1.0, 100%)
Patch Gaussian(fixed, 150, 2.0, 100%)
Patch Gaussian(fixed, 200, 0.2, 100%)
Patch Gaussian(fixed, 200, 0.5, 100%)
Patch Gaussian(fixed, 200, 0.8, 100%)

Patch Gaussian(fixed, 200, 1.0, 100%)
Patch Gaussian(fixed, 200, 2.0, 100%)
Patch Gaussian(fixed, 250, 0.2, 100%)
Patch Gaussian(fixed, 250, 0.5, 100%)
Patch Gaussian(fixed, 250, 0.8, 100%)
Patch Gaussian(fixed, 250, 1.0, 100%)
Patch Gaussian(fixed, 250, 2.0, 100%)
Patch Gaussian(fixed, 300, 0.2, 100%)
Patch Gaussian(fixed, 300, 0.5, 100%)
Patch Gaussian(fixed, 300, 0.8, 100%)
Patch Gaussian(fixed, 300, 1.0, 100%)
Patch Gaussian(fixed, 300, 2.0, 100%)
Patch Gaussian(fixed, 350, 0.2, 100%)
Patch Gaussian(fixed, 350, 0.5, 100%)
Patch Gaussian(fixed, 350, 0.8, 100%)
Patch Gaussian(fixed, 350, 1.0, 100%)
Patch Gaussian(fixed, 350, 2.0, 100%)
Patch Gaussian(fixed, 400, 0.2, 100%)
Patch Gaussian(fixed, 400, 0.5, 100%)
Patch Gaussian(fixed, 400, 0.8, 100%)
Patch Gaussian(fixed, 400, 1.0, 100%)
Patch Gaussian(fixed, 400, 2.0, 100%)
Posterize(0, 100%)
Posterize(1, 100%)
Posterize(2, 100%)
Posterize(3, 100%)
Posterize(4, 100%)
Posterize(5, 100%)
Posterize(6, 100%)
Posterize(7, 100%)
Random Erasing(variable, 448, 100%)
Random Erasing(fixed, 120, 100%)
Random Erasing(fixed, 150, 100%)
Random Erasing(fixed, 180, 100%)
Random Erasing(fixed, 30, 100%)
Random Erasing(fixed, 60, 100%)
Random Erasing(fixed, 90, 100%)
Rotate(variable, 0deg, 100%)
Rotate(variable, 10deg, 100%)
Rotate(variable, 15deg, 100%)
Rotate(variable, 20deg, 100%)
Rotate(variable, 25deg, 100%)
Rotate(variable, 30deg, 100%)
Rotate(variable, 5deg, 100%)
Sharpness(0.1, 100%)
Sharpness(0.2, 100%)
Sharpness(0.3, 100%)
Sharpness(0.4, 100%)
Sharpness(0.5, 100%)
Sharpness(0.6, 100%)
Sharpness(0.7, 100%)
ShearX(variable, 0.1, 100%)
ShearX(variable, 0.2, 100%)
ShearX(variable, 0.3, 100%)
ShearX(variable, 0.4, 100%)
ShearX(variable, 0.5, 100%)
Solarize(0, 100%)
Solarize(100, 100%)
Solarize(150, 100%)
Solarize(200, 100%)
Solarize(250, 100%)
Solarize(50, 100%)
Solarize Add(-002, 000, 100%)
Solarize Add(-002, 050, 100%)
Solarize Add(-002, 100, 100%)
Solarize Add(-002, 150, 100%)
Solarize Add(-002, 200, 100%)
Solarize Add(-002, 250, 100%)
Solarize Add(-027, 000, 100%)
Solarize Add(-027, 050, 100%)
Solarize Add(-027, 100, 100%)
Solarize Add(-027, 150, 100%)
Solarize Add(-027, 200, 100%)
Solarize Add(-027, 250, 100%)
Solarize Add(-052, 000, 100%)
Solarize Add(-052, 050, 100%)
Solarize Add(-052, 100, 100%)
Solarize Add(-052, 150, 100%)
Solarize Add(-052, 200, 100%)
Solarize Add(-052, 250, 100%)
Solarize Add(-077, 000, 100%)
Solarize Add(-077, 050, 100%)
Solarize Add(-077, 100, 100%)
Solarize Add(-077, 150, 100%)
Solarize Add(-077, 200, 100%)
Solarize Add(-077, 250, 100%)
Solarize Add(-102, 000, 100%)
Solarize Add(-102, 050, 100%)
Solarize Add(-102, 100, 100%)
Solarize Add(-102, 150, 100%)
Solarize Add(-102, 200, 100%)
Solarize Add(-102, 250, 100%)
Solarize Add(-127, 000, 100%)
Solarize Add(-127, 050, 100%)

Solarize Add(-127, 100, 100%)
Solarize Add(-127, 150, 100%)
Solarize Add(-127, 200, 100%)
Solarize Add(-127, 250, 100%)
Solarize Add(0023, 050, 100%)
Solarize Add(0023, 100, 100%)
Solarize Add(0023, 150, 100%)
Solarize Add(0023, 200, 100%)
Solarize Add(0023, 250, 100%)
Solarize Add(0048, 050, 100%)
Solarize Add(0048, 100, 100%)
Solarize Add(0048, 150, 100%)
Solarize Add(0048, 200, 100%)
Solarize Add(0048, 250, 100%)
Solarize Add(0073, 100, 100%)
Solarize Add(0073, 150, 100%)
Solarize Add(0073, 200, 100%)
Solarize Add(0073, 250, 100%)
Solarize Add(0098, 100, 100%)
Solarize Add(0098, 150, 100%)
Solarize Add(0098, 200, 100%)
Solarize Add(0098, 250, 100%)
Solarize Add(0123, 150, 100%)
Solarize Add(0123, 200, 100%)
Solarize Add(0123, 250, 100%)
TranslateX(0, 100%)
TranslateX(10, 100%)
TranslateX(20, 100%)
TranslateX(30, 100%)
TranslateX(40, 100%)
TranslateX(50, 100%)
TranslateX(60, 100%)
TranslateX(70, 100%)
TranslateX(80, 100%)
TranslateX(90, 100%)
RandomMultiAug(1, 1, 0)
RandomMultiAug(1, 1, 1)
RandomMultiAug(1, 3, 0)
RandomMultiAug(1, 3, 1)
RandomMultiAug(1, 7, 0)
RandomMultiAug(1, 7, 1)
RandomMultiAug(1, 10, 0)
RandomMultiAug(1, 10, 1)
RandomMultiAug(1, 15, 0)
RandomMultiAug(1, 15, 1)
RandomMultiAug(1, 20, 0)
RandomMultiAug(1, 20, 1)
RandomMultiAug(2, 1, 0)
RandomMultiAug(2, 1, 1)
RandomMultiAug(2, 3, 0)
RandomMultiAug(2, 3, 1)
RandomMultiAug(2, 7, 0)
RandomMultiAug(2, 7, 1)
RandomMultiAug(2, 10, 0)
RandomMultiAug(2, 10, 1)
RandomMultiAug(2, 15, 0)
RandomMultiAug(2, 15, 1)
RandomMultiAug(2, 20, 0)
RandomMultiAug(2, 20, 1)
RandomMultiAug(3, 1, 0)
RandomMultiAug(3, 1, 1)
RandomMultiAug(3, 3, 0)
RandomMultiAug(3, 3, 1)
RandomMultiAug(3, 7, 0)
RandomMultiAug(3, 7, 1)
RandomMultiAug(3, 10, 0)
RandomMultiAug(3, 10, 1)
RandomMultiAug(3, 15, 0)
RandomMultiAug(3, 15, 1)
RandomMultiAug(3, 20, 0)
RandomMultiAug(3, 20, 1)
RandomMultiAug(4, 1, 0)
RandomMultiAug(4, 1, 1)
RandomMultiAug(4, 3, 0)
RandomMultiAug(4, 3, 1)
RandomMultiAug(4, 7, 0)
RandomMultiAug(4, 7, 1)
RandomMultiAug(4, 10, 0)
RandomMultiAug(4, 10, 1)
RandomMultiAug(4, 15, 0)
RandomMultiAug(4, 15, 1)
RandomMultiAug(4, 20, 0)
RandomMultiAug(4, 20, 1)
RandomMultiAug(5, 1, 0)
RandomMultiAug(5, 1, 1)
RandomMultiAug(5, 3, 0)
RandomMultiAug(5, 3, 1)
RandomMultiAug(5, 7, 0)
RandomMultiAug(5, 7, 1)
RandomMultiAug(5, 10, 0)
RandomMultiAug(5, 15, 0)
RandomMultiAug(5, 15, 1)
RandomMultiAug(5, 20, 0)
RandomMultiAug(5, 20, 1)

Each augmentation was applied with a certain probability (given as a percentage in the label). Each time an image was pulled for training, the given image was augmented with that probability.

## C    Error analysis

All of the CIFAR-10 experiments were repeated with 10 different initialization. In most cases, the resulting standard error on the mean (SEM) is too small to show as error bars on plots. The error on each measurement is given in the full results (see Sec. G).

Affinity and Switch-off Lift both were computed from differences between runs that share the same initialization. For Affinity, the same trained model was used for inference on clean validation data and on augmented validation data. Thus, the variance of Affinity for the clean baseline is not independent of the variance of Affinity for a given augmentation. The difference between the augmentation case and the clean baseline case was taken on a per-experiment basis (for each initialization of the clean baseline model) before the error was computed.

In the switching experiments, the final training *without* augmentation was completed starting from a given checkpoint in the model that was trained *with* augmentation. Thus, each switching experiment shared an initialization with an experiment that had no switching. Again, in this case the difference was taken on a per-experiment basis before the error (based on the standard deviation) was computed.

All ImageNet experiments shown are with one initialization. Thus, there are not statistics from which to analyze the error.

## D    Switching off augmentations

For CIFAR-10, switching times were tested in increments of approximately 5k steps between $\sim 25$k and $\sim 75$k steps. The best point for switching was determined by the final validation accuracy.

On ImageNet, we tested turning augmentation off at 50, 60, 70, and 80 epochs. Total training took 90 epochs. The best point for switching was determined by the final test accuracy.

The Switch-off Lift was derived from the experiment at the best switch-off point for each augmentation.

For CIFAR-10, there are some augmentations where the validation accuracy was best at 25k, which means that further testing is needed to find if the actual optimum switch off point is lower or if the best case is to not train at all with the given augmentation. Some of the best augmentations have a small negative Switch-off Lift, indicating that it is better to train the entire time with the given augmentations. For completeness we have listed the constant and optimal switching performance for all augmentations studied in the table below. The error is the standard error on the mean of the difference between the two over ten runs.

| Aug name long | Constant Aug | Turn Aug Off | Switching Lift Err |
|---|---|---|---|
| Clean | 0.90 | | |
| Blur(100%) | 0.54 | 0.89 | 1.1e-02 |
| Blur(50%) | 0.89 | 0.90 | 1.6e-03 |
| Crop(4, 25%) | 0.93 | 0.93 | 2.7e-04 |
| Crop(4, 50%) | 0.93 | 0.93 | 2.5e-04 |
| Crop(4, 75%) | 0.93 | 0.93 | 3.1e-04 |
| Crop(4,100%) | 0.93 | 0.93 | 3.3e-04 |
| Cutout(16, 100%) | 0.92 | 0.92 | 5.1e-04 |
| Cutout(16, 25%) | 0.92 | 0.92 | 2.5e-04 |
| Cutout(16, 50%) | 0.92 | 0.92 | 6.8e-04 |
| Cutout(16, 75%) | 0.92 | 0.92 | 6.5e-04 |
| Equalize(100%) | 0.81 | 0.90 | 4.8e-03 |
| Equalize(50%) | 0.89 | 0.90 | 1.5e-03 |
| FlipLR(100%) | 0.89 | 0.92 | 1.4e-03 |
| FlipLR(25%) | 0.93 | 0.93 | 3.0e-04 |
| FlipLR(50%) | 0.93 | 0.93 | 1.9e-04 |
| FlipLR(50%) + Crop(4, 100%) + Cutout(16, 100%)... | 0.95 | 0.95 | 4.0e-04 |
| FlipLR(50%) + Crop(4, 100%) + Cutout(16, 100%)... | 0.95 | 0.95 | 4.4e-04 |
| FlipLR(50%) + Crop(4,100%) | 0.95 | 0.95 | 4.8e-04 |
| FlipLR(75%) | 0.93 | 0.93 | 2.0e-04 |
| FlipUD(100%) | 0.40 | 0.90 | 2.1e-03 |

| | | | |
|---|---|---|---|
| FlipUD(25%) | 0.90 | 0.91 | 6.5e-04 |
| FlipUD(50%) | 0.90 | 0.91 | 7.3e-04 |
| FlipUD(75%) | 0.89 | 0.91 | 1.0e-03 |
| Invert(100%) | 0.58 | 0.90 | 4.3e-03 |
| Invert(50%) | 0.88 | 0.90 | 2.0e-03 |
| Rotate(fixed, 15deg, 50%) | 0.93 | 0.93 | 2.9e-04 |
| Rotate(fixed, 20deg, 100%) | 0.85 | 0.92 | 2.1e-03 |
| Rotate(fixed, 20deg, 50%) | 0.93 | 0.93 | 3.0e-04 |
| Rotate(fixed, 20deg, 75%) | 0.92 | 0.93 | 3.1e-04 |
| Rotate(fixed, 45deg, 50%) | 0.92 | 0.92 | 3.7e-04 |
| Rotate(fixed, 5deg, 75%) | 0.92 | 0.92 | 1.6e-04 |
| Rotate(square, 100%) | 0.90 | 0.92 | 9.5e-04 |
| Rotate(square, 50%) | 0.91 | 0.92 | 7.5e-04 |
| Rotate(variable, 20deg, 100%) | 0.93 | 0.93 | 3.2e-04 |
| Rotate(variable, 20deg, 50%) | 0.93 | 0.93 | 4.8e-04 |
| Rotate(variable, 20deg, 75%) | 0.93 | 0.93 | 2.5e-04 |
| Rotate(variable, 45, 100%) | 0.92 | 0.93 | 1.7e-03 |
| Rotate(variable, 5deg, 100%) | 0.92 | 0.92 | 2.0e-04 |
| Rotate(variable, 5deg, 50%) | 0.92 | 0.92 | 2.7e-04 |
| Rotate(variable, 5deg, 75%) | 0.92 | 0.92 | 1.6e-04 |
| Rotate(variable, 60deg, 100%) | 0.92 | 0.93 | 5.3e-04 |
| ShearX(fixed ,0.1, 100%) | 0.91 | 0.92 | 8.0e-04 |
| ShearX(fixed, 0.1, 50%) | 0.92 | 0.92 | 3.2e-04 |
| ShearX(fixed, 0.1, 75%) | 0.92 | 0.92 | 3.4e-04 |
| ShearX(fixed, 0.3, 100%) | 0.90 | 0.91 | 8.1e-04 |
| ShearX(fixed, 0.3, 50%) | 0.92 | 0.92 | 4.2e-04 |
| ShearX(fixed, 0.3, 75%) | 0.92 | 0.92 | 2.6e-04 |
| ShearX(variable, 0.1, 100%) | 0.92 | 0.92 | 3.2e-04 |
| ShearX(variable, 0.1, 50%) | 0.92 | 0.92 | 3.0e-04 |
| ShearX(variable, 0.1, 75%) | 0.92 | 0.92 | 2.9e-04 |
| ShearX(variable, 0.3, 100%) | 0.92 | 0.92 | 3.6e-04 |
| ShearX(variable, 0.3, 50%) | 0.92 | 0.92 | 2.4e-04 |
| ShearX(variable, 0.3, 75%) | 0.92 | 0.92 | 3.4e-04 |
| FlipLR(25%) + Crop(4,25%) + Cutout(16,25%) | 0.95 | 0.95 | 2.3e-04 |
| FlipLR(25%) + Crop(4,25%) + Cutout(16,50%) | 0.95 | 0.95 | 3.7e-04 |
| FlipLR(25%) + Crop(4,25%) + Cutout(16,75%) | 0.95 | 0.95 | 3.5e-04 |
| FlipLR(25%) + Crop(4,25%) + Cutout(16,100%) | 0.95 | 0.95 | 4.6e-04 |
| FlipLR(25%) + Crop(4,50%) + Cutout(16,25%) | 0.95 | 0.95 | 3.7e-04 |
| FlipLR(25%) + Crop(4,50%) + Cutout(16,50%) | 0.95 | 0.95 | 4.7e-04 |
| FlipLR(25%) + Crop(4,50%) + Cutout(16,75%) | 0.95 | 0.95 | 3.6e-04 |
| FlipLR(25%) + Crop(4,50%) + Cutout(16,100%) | 0.95 | 0.95 | 3.0e-04 |
| FlipLR(25%) + Crop(4,75%) + Cutout(16,25%) | 0.95 | 0.95 | 2.4e-04 |
| FlipLR(25%) + Crop(4,75%) + Cutout(16,50%) | 0.95 | 0.95 | 3.6e-04 |
| FlipLR(25%) + Crop(4,75%) + Cutout(16,75%) | 0.95 | 0.95 | 3.3e-04 |
| FlipLR(25%) + Crop(4,75%) + Cutout(16,100%) | 0.95 | 0.95 | 5.4e-04 |
| FlipLR(25%) + Crop(4,100%) + Cutout(16,25%) | 0.95 | 0.95 | 3.2e-04 |
| FlipLR(25%) + Crop(4,100%) + Cutout(16,50%) | 0.95 | 0.95 | 3.3e-04 |
| FlipLR(25%) + Crop(4,100%) + Cutout(16,75%) | 0.95 | 0.95 | 2.5e-04 |
| FlipLR(25%) + Crop(4,100%) + Cutout(16,100%) | 0.95 | 0.95 | 3.4e-04 |
| FlipLR(50%) + Crop(4,25%) + Cutout(16,25%) | 0.95 | 0.95 | 2.3e-04 |
| FlipLR(50%) + Crop(4,25%) + Cutout(16,50%) | 0.95 | 0.95 | 2.2e-04 |
| FlipLR(50%) + Crop(4,25%) + Cutout(16,75%) | 0.95 | 0.95 | 4.7e-04 |
| FlipLR(50%) + Crop(4,25%) + Cutout(16,100%) | 0.95 | 0.95 | 3.5e-04 |
| FlipLR(50%) + Crop(4,50%) + Cutout(16,25%) | 0.95 | 0.95 | 3.1e-04 |
| FlipLR(50%) + Crop(4,50%) + Cutout(16,50%) | 0.95 | 0.95 | 3.8e-04 |
| FlipLR(50%) + Crop(4,50%) + Cutout(16,75%) | 0.95 | 0.95 | 3.7e-04 |
| FlipLR(50%) + Crop(4,50%) + Cutout(16,100%) | 0.95 | 0.95 | 5.6e-04 |
| FlipLR(50%) + Crop(4,75%) + Cutout(16,25%) | 0.95 | 0.95 | 3.2e-04 |
| FlipLR(50%) + Crop(4,75%) + Cutout(16,50%) | 0.95 | 0.95 | 4.3e-04 |
| FlipLR(50%) + Crop(4,75%) + Cutout(16,75%) | 0.96 | 0.95 | 6.1e-04 |
| FlipLR(50%) + Crop(4,75%) + Cutout(16,100%) | 0.95 | 0.95 | 4.2e-04 |
| FlipLR(50%) + Crop(4,100%) + Cutout(16,25%) | 0.95 | 0.95 | 3.7e-04 |
| FlipLR(50%) + Crop(4,100%) + Cutout(16,50%) | 0.95 | 0.96 | 3.6e-04 |
| FlipLR(50%) + Crop(4,100%) + Cutout(16,75%) | 0.95 | 0.95 | 2.8e-04 |
| FlipLR(50%) + Crop(4,100%) + Cutout(16,100%) | 0.96 | 0.96 | 5.0e-04 |

| | | | |
|---|---|---|---|
| FlipLR(75%) + Crop(4,25%) + Cutout(16,25%) | 0.95 | 0.95 | 2.7e-04 |
| FlipLR(75%) + Crop(4,25%) + Cutout(16,50%) | 0.95 | 0.95 | 5.0e-04 |
| FlipLR(75%) + Crop(4,25%) + Cutout(16,75%) | 0.95 | 0.95 | 3.9e-04 |
| FlipLR(75%) + Crop(4,25%) + Cutout(16,100%) | 0.95 | 0.95 | 3.3e-04 |
| FlipLR(75%) + Crop(4,50%) + Cutout(16,25%) | 0.95 | 0.95 | 2.6e-04 |
| FlipLR(75%) + Crop(4,50%) + Cutout(16,50%) | 0.95 | 0.95 | 3.7e-04 |
| FlipLR(75%) + Crop(4,50%) + Cutout(16,75%) | 0.95 | 0.95 | 3.4e-04 |
| FlipLR(75%) + Crop(4,50%) + Cutout(16,100%) | 0.95 | 0.95 | 3.5e-04 |
| FlipLR(75%) + Crop(4,75%) + Cutout(16,25%) | 0.95 | 0.95 | 3.6e-04 |
| FlipLR(75%) + Crop(4,75%) + Cutout(16,50%) | 0.95 | 0.95 | 3.6e-04 |
| FlipLR(75%) + Crop(4,75%) + Cutout(16,75%) | 0.95 | 0.95 | 3.4e-04 |
| FlipLR(75%) + Crop(4,75%) + Cutout(16,100%) | 0.95 | 0.95 | 3.0e-04 |
| FlipLR(75%) + Crop(4,100%) + Cutout(16,25%) | 0.95 | 0.95 | 4.3e-04 |
| FlipLR(75%) + Crop(4,100%) + Cutout(16,50%) | 0.95 | 0.95 | 2.7e-04 |
| FlipLR(75%) + Crop(4,100%) + Cutout(16,75%) | 0.95 | 0.95 | 4.5e-04 |
| FlipLR(75%) + Crop(4,100%) + Cutout(16,100%) | 0.95 | 0.95 | 3.3e-04 |
| FlipLR(100%) + Crop(4,25%) + Cutout(16,25%) | 0.93 | 0.94 | 7.9e-04 |
| FlipLR(100%) + Crop(4,25%) + Cutout(16,50%) | 0.94 | 0.94 | 4.7e-04 |
| FlipLR(100%) + Crop(4,25%) + Cutout(16,75%) | 0.94 | 0.94 | 7.1e-04 |
| FlipLR(100%) + Crop(4,25%) + Cutout(16,100%) | 0.94 | 0.94 | 5.3e-04 |
| FlipLR(100%) + Crop(4,50%) + Cutout(16,25%) | 0.94 | 0.94 | 5.8e-04 |
| FlipLR(100%) + Crop(4,50%) + Cutout(16,50%) | 0.94 | 0.94 | 5.2e-04 |
| FlipLR(100%) + Crop(4,50%) + Cutout(16,75%) | 0.94 | 0.94 | 5.0e-04 |
| FlipLR(100%) + Crop(4,50%) + Cutout(16,100%) | 0.94 | 0.94 | 5.8e-04 |
| FlipLR(100%) + Crop(4,75%) + Cutout(16,25%) | 0.94 | 0.94 | 5.0e-04 |
| FlipLR(100%) + Crop(4,75%) + Cutout(16,50%) | 0.94 | 0.94 | 7.5e-04 |
| FlipLR(100%) + Crop(4,75%) + Cutout(16,75%) | 0.94 | 0.95 | 4.4e-04 |
| FlipLR(100%) + Crop(4,75%) + Cutout(16,100%) | 0.94 | 0.95 | 6.6e-04 |
| FlipLR(100%) + Crop(4,100%) + Cutout(16,25%) | 0.94 | 0.94 | 5.3e-04 |
| FlipLR(100%) + Crop(4,100%) + Cutout(16,50%) | 0.94 | 0.94 | 4.7e-04 |
| FlipLR(100%) + Crop(4,100%) + Cutout(16,75%) | 0.94 | 0.95 | 6.5e-04 |
| FlipLR(100%) + Crop(4,100%) + Cutout(16,100%) | 0.94 | 0.95 | 4.9e-04 |
| Rotate(fixed, 5deg, 10%) | 0.92 | 0.92 | 1.9e-04 |
| Rotate(fixed, 5deg, 20%) | 0.92 | 0.92 | 3.6e-04 |
| Rotate(fixed, 5deg, 30%) | 0.92 | 0.92 | 3.1e-04 |
| Rotate(fixed, 5deg, 40%) | 0.92 | 0.92 | 1.9e-04 |
| Rotate(fixed, 5deg, 50%) | 0.92 | 0.92 | 3.0e-04 |
| Rotate(fixed, 5deg, 60%) | 0.92 | 0.92 | 2.6e-04 |
| Rotate(fixed, 5deg, 70%) | 0.92 | 0.92 | 2.9e-04 |
| Rotate(fixed, 5deg, 80%) | 0.92 | 0.92 | 2.3e-04 |
| Rotate(fixed, 5deg, 90%) | 0.92 | 0.92 | 3.4e-04 |
| Rotate(fixed, 5deg, 100%) | 0.92 | 0.92 | 4.5e-04 |
| Rotate(fixed, 60deg, 10%) | 0.91 | 0.91 | 4.3e-04 |
| Rotate(fixed, 60deg, 20%) | 0.91 | 0.91 | 6.0e-04 |
| Rotate(fixed, 60deg, 30%) | 0.91 | 0.92 | 4.7e-04 |
| Rotate(fixed, 60deg, 40%) | 0.91 | 0.92 | 5.0e-04 |
| Rotate(fixed, 60deg, 50%) | 0.92 | 0.92 | 1.9e-03 |
| Rotate(fixed, 60deg, 60%) | 0.91 | 0.91 | 5.3e-04 |
| Rotate(fixed, 60deg, 70%) | 0.91 | 0.92 | 1.4e-03 |
| Rotate(fixed, 60deg, 80%) | 0.91 | 0.92 | 1.9e-03 |
| Rotate(fixed, 60deg, 90%) | 0.90 | 0.91 | 1.5e-03 |
| Rotate(fixed, 60deg, 100%) | 0.41 | 0.91 | 3.9e-03 |
| PatchGaussian(fixed,4, 0.1, 100%) | 0.90 | 0.90 | 4.3e-04 |
| PatchGaussian(fixed,4, 0.2, 100%) | 0.90 | 0.90 | 3.5e-04 |
| PatchGaussian(fixed,4, 0.3, 100%) | 0.90 | 0.90 | 3.0e-04 |
| PatchGaussian(fixed,4, 0.5, 100%) | 0.91 | 0.91 | 2.6e-04 |
| PatchGaussian(fixed,4, 0.8, 100%) | 0.91 | 0.91 | 3.6e-04 |
| PatchGaussian(fixed,4, 1.0, 100%) | 0.91 | 0.91 | 2.8e-04 |
| PatchGaussian(fixed,4, 1.5, 100%) | 0.91 | 0.91 | 4.2e-04 |
| PatchGaussian(fixed,4, 2.0, 100%) | 0.91 | 0.91 | 4.4e-04 |
| PatchGaussian(fixed,8, 0.1, 100%) | 0.90 | 0.90 | 7.6e-04 |
| PatchGaussian(fixed,8, 0.2, 100%) | 0.91 | 0.91 | 2.8e-04 |
| PatchGaussian(fixed,8, 0.3, 100%) | 0.91 | 0.91 | 5.5e-04 |
| PatchGaussian(fixed,8, 0.5, 100%) | 0.91 | 0.91 | 3.0e-04 |
| PatchGaussian(fixed,8, 0.8, 100%) | 0.92 | 0.91 | 3.6e-04 |

| | | | |
|---|---|---|---|
| PatchGaussian(fixed,8, 1.0, 100%) | 0.92 | 0.91 | 4.6e-04 |
| PatchGaussian(fixed,8, 1.5, 100%) | 0.92 | 0.92 | 4.6e-04 |
| PatchGaussian(fixed,8, 2.0, 100%) | 0.92 | 0.92 | 5.8e-04 |
| PatchGaussian(fixed,12, 0.1, 100%) | 0.90 | 0.90 | 2.3e-04 |
| PatchGaussian(fixed,12, 0.2, 100%) | 0.91 | 0.91 | 4.3e-04 |
| PatchGaussian(fixed,12, 0.3, 100%) | 0.91 | 0.91 | 5.2e-04 |
| PatchGaussian(fixed,12, 0.5, 100%) | 0.91 | 0.91 | 4.4e-04 |
| PatchGaussian(fixed,12, 0.8, 100%) | 0.91 | 0.91 | 6.7e-04 |
| PatchGaussian(fixed,12, 1.0, 100%) | 0.92 | 0.92 | 4.9e-04 |
| PatchGaussian(fixed,12, 1.5, 100%) | 0.92 | 0.91 | 4.0e-04 |
| PatchGaussian(fixed,12, 2.0, 100%) | 0.92 | 0.92 | 7.3e-04 |
| PatchGaussian(fixed,16, 0.1, 100%) | 0.90 | 0.90 | 5.1e-04 |
| PatchGaussian(fixed,16, 0.2, 100%) | 0.91 | 0.90 | 6.1e-04 |
| PatchGaussian(fixed,16, 0.3, 100%) | 0.91 | 0.90 | 7.4e-04 |
| PatchGaussian(fixed,16, 0.5, 100%) | 0.91 | 0.90 | 7.7e-04 |
| PatchGaussian(fixed,16, 0.8, 100%) | 0.90 | 0.91 | 1.4e-03 |
| PatchGaussian(fixed,16, 1.0, 100%) | 0.90 | 0.91 | 1.3e-03 |
| PatchGaussian(fixed,16, 1.5, 100%) | 0.90 | 0.90 | 7.5e-04 |
| PatchGaussian(fixed,16, 2.0, 100%) | 0.90 | 0.91 | 1.6e-03 |
| PatchGaussian(fixed,20, 0.1, 100%) | 0.90 | 0.90 | 1.3e-03 |
| PatchGaussian(fixed,20, 0.2, 100%) | 0.89 | 0.90 | 1.1e-03 |
| PatchGaussian(fixed,20, 0.3, 100%) | 0.89 | 0.90 | 1.6e-03 |
| PatchGaussian(fixed,20, 0.5, 100%) | 0.89 | 0.90 | 1.6e-03 |
| PatchGaussian(fixed,20, 0.8, 100%) | 0.88 | 0.90 | 1.8e-03 |
| PatchGaussian(fixed,20, 1.0, 100%) | 0.88 | 0.91 | 1.1e-03 |
| PatchGaussian(fixed,20, 1.5, 100%) | 0.87 | 0.90 | 1.8e-03 |
| PatchGaussian(fixed,20, 2.0, 100%) | 0.88 | 0.90 | 1.4e-03 |
| PatchGaussian(fixed,24, 0.1, 100%) | 0.89 | 0.90 | 9.7e-04 |
| PatchGaussian(fixed,24, 0.2, 100%) | 0.88 | 0.90 | 1.4e-03 |
| PatchGaussian(fixed,24, 0.3, 100%) | 0.87 | 0.90 | 1.3e-03 |
| PatchGaussian(fixed,24, 0.5, 100%) | 0.86 | 0.90 | 2.0e-03 |
| PatchGaussian(fixed,24, 0.8, 100%) | 0.84 | 0.90 | 1.6e-03 |
| PatchGaussian(fixed,24, 1.0, 100%) | 0.84 | 0.90 | 1.7e-03 |
| PatchGaussian(fixed,24, 1.5, 100%) | 0.83 | 0.90 | 2.4e-03 |
| PatchGaussian(fixed,24, 2.0, 100%) | 0.82 | 0.90 | 9.6e-04 |
| PatchGaussian(fixed,28, 0.1, 100%) | 0.88 | 0.90 | 1.0e-03 |
| PatchGaussian(fixed,28, 0.2, 100%) | 0.85 | 0.90 | 1.4e-03 |
| PatchGaussian(fixed,28, 0.3, 100%) | 0.83 | 0.90 | 1.8e-03 |
| PatchGaussian(fixed,28, 0.5, 100%) | 0.81 | 0.90 | 1.7e-03 |
| PatchGaussian(fixed,28, 0.8, 100%) | 0.79 | 0.90 | 2.4e-03 |
| PatchGaussian(fixed,28, 1.0, 100%) | 0.77 | 0.89 | 1.7e-03 |
| PatchGaussian(fixed,28, 1.5, 100%) | 0.74 | 0.89 | 2.4e-03 |
| PatchGaussian(fixed,28, 2.0, 100%) | 0.72 | 0.90 | 3.3e-03 |
| PatchGaussian(fixed,32, 0.1, 100%) | 0.88 | 0.90 | 1.8e-03 |
| PatchGaussian(fixed,32, 0.2, 100%) | 0.85 | 0.90 | 1.6e-03 |
| PatchGaussian(fixed,32, 0.3, 100%) | 0.82 | 0.90 | 1.8e-03 |
| PatchGaussian(fixed,32, 0.5, 100%) | 0.79 | 0.90 | 1.8e-03 |
| PatchGaussian(fixed,32, 0.8, 100%) | 0.79 | 0.89 | 2.5e-03 |
| PatchGaussian(fixed,32, 1.0, 100%) | 0.79 | 0.89 | 2.0e-03 |
| PatchGaussian(fixed,32, 1.5, 100%) | 0.81 | 0.89 | 1.5e-03 |
| PatchGaussian(fixed,32, 2.0, 100%) | 0.81 | 0.89 | 1.7e-03 |

## E  DIVERSITY METRICS

We computed three possible diversity metrics, shown in Fig. 9: Entropy, Final Training Loss, Training Steps to Accuracy Threshold. The entropy was calculated only for augmentations that have a discrete stochasticity (such as `Rotate(fixed)` and not for augmentations that have a continuous variation (such as `Rotate(variable)` or `PatchGaussian`). Final Training Loss is the batch statistic at the last step of training. For CIFAR-10 experiments, this was averaged across the 10 initializations. For ImageNet, it was averaged over the last 10 steps of training. Training Steps to Accuracy Threshold is the number of training steps at which the training accuracy first hits a threshold of 97%. A few of the tested augmentation (extreme versions of `PatchGaussian`) did not reach this threshold in the given time and that column is left blank in the full results.

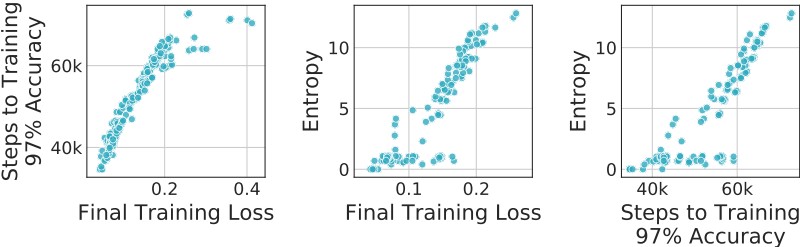

Figure 9: CIFAR-10: Three different diversity metrics are strongly correlated for high entropy augmentations. Here, the entropy is calculated only for discrete augmentations.

Entropy is unique in that it is independent of the model or dataset and it is a counting of states. However, it is difficult to compare between discrete and continuously-varying transforms and it is not clear how proper it is to compare even across different types of transforms.

Final Training Loss and Training Steps to Accuracy Threshold correlate well across the tested transforms. Entropy is highly correlated to these measures for `PatchGaussian` and versions of `FlipLR`, `Crop`, and `Cutout` where only probabilities are varying. For `Rotate` and `Shear` where magnitudes are varying as well, the correlation between Entropy and the other two measures is less clear.

One shortcoming of the loss or training step based Diversity measures is that they require fully training a new model for each augmentation policy. To ameliorate this computational cost, we also consider using the training loss, but for models trained for a significantly shorter time. In Figure 10 and Figure 11. We consider CIFAR-10 models trained for 4.5% and ImageNet models trained for 12% of the full training time. We find good correlation between this short time measure and the training loss computed at the end of training. With this alternate computationally less-expensive measure we find that 98.9% of pairs of CIFAR-10 models and 9.66% of ImageNet models satisfy Inequality 3.

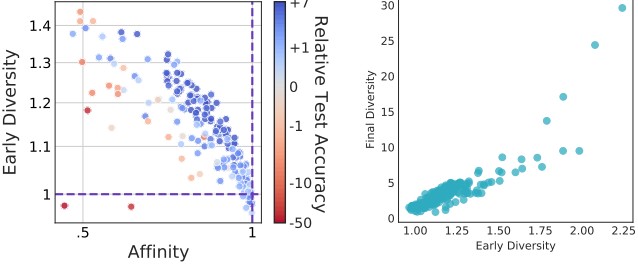

Figure 10: CIFAR-10: (Left) Diversity as the average loss over the first 10 epochs (out of 222) epochs of training (10 measurements, one per epoch). This represents an 96.5% reduction in computational cost. 98.9% of pairs satisfy Inequality 3. (Right) Early Diversity compared to Diversity computed for full training, this represents a Spearman correlation of 0.87

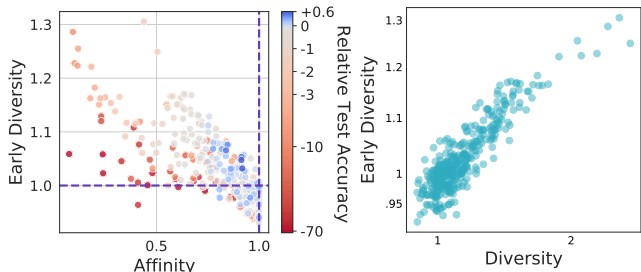

Figure 11: ImageNet: (Left) Diversity as the average loss over the first 13000 (out of 112000) steps of training (5 measurements, one per epoch), this represents an 88% reduction in computational cost. 96.6% of pairs satisfy Inequality 3. (Right) Early Diversity compared to Diversity computed for full training. This represents a Spearman correlation of 0.82.

## F  COMPARING AFFINITY TO OTHER RELATED MEASURES

We gain confidence in the Affinity measure by comparing it to other potential model-dependant measures of distribution shift. In Fig 12, we show the correlation between Affinity and these two measures: the mean log likelihood of augmented test images(Grathwohl et al., 2019) (labeled as "logsumexp(logits)") and the Watanabe–Akaike information criterion (labeled as "WAIC") (Watanabe, 2010).

Like Affinity, these other two measures indicate how well a model trained on clean data comprehends augmented data.

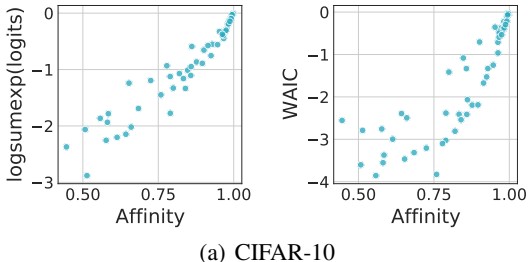

(a) CIFAR-10

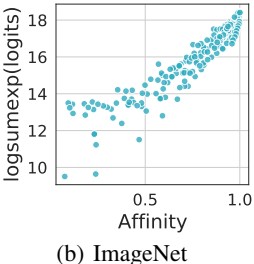

(b) ImageNet

Figure 12: Affinity correlates with two other measures of how augmented images are related to a trained model's distribution: `logsumexp` of the logits (left, for CIFAR-10, and right, for ImageNet) is the mean log likelihood for the image. WAIC (middle, for CIFAR-10) corrects for a possible bias in that estimate. In all three plots, numbers are referenced to the clean baseline, which is assigned a value of 0.

## G  FULL RESULTS

The plotted data for CIFAR-10 and ImageNet are given in .csv files uploaded at `https://storage.googleapis.com/public_research_data/augmentation/data.zip`. In these .csv files, blank cells generally indicate that a given experiment (such as switching) was not done for the specified augmentation. In the case of the training accuracy threshold as a proxy for diversity, a blank cell indicates that for the given augmentation, the training accuracy did not reach the specified threshold during training.

## H  MODEL ROBUSTNESS ON THE AFFINITY-DIVERSITY PLANE

To shed light on whether Affinity and Diversity inform model robustness we evaluated our ImageNet and CIFAR-10 models on ImageNet-V2 and CIFAR-10.1 (Recht et al., 2019; 2018) as well as on CIFAR-10-C (Hendrycks & Dietterich, 2019). The former two datasets attempt to recreate the CIFAR-10 and ImageNet test sets as closely as possible, however model performance typically suffers on these sets indicating some degree of

distribution shift. CIFAR-10-C is a dataset designed to test model robustness to a variety of image corruptions. We display accuracy for each individual corruption in Figure 16 and the overall CIFAR-10-C score in Figure 15. We find that for both ImageNet-V2 and CIFAR-10.1, and the robustness gap (difference between clean and robust accuracy) show consistent trends with higher diversity and higher affinity augmentations leading to more robust performance. For some of the CIFAR-10-C corruptions we see a similar trend. For other augmentations, (e.g. Gaussian noise, impulse noise, speckle and shot noise) we see that models trained with patch Gaussian augmentations (red-circles) outperform this general trend. This is perhaps consistent with the picture that models trained with augmentations well paired with a specific corruption perform well (Yin et al., 2019; Anonymous, 2021).

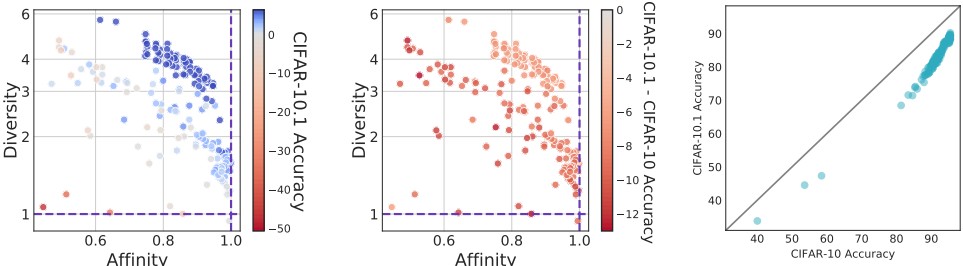

Figure 13: (Left) Augmented model performance on CIFAR-10.1 relative to a clean baseline (acc 80.1). We see that robust accuracy is higher for models trained with more diverse higher affinity augmentation policies. (Middle) Gap in performance between CIFAR-10.1 and CIFAR-10 accuracy. (Right) CIFAR-10.1 accuracy versus CIFAR-10 accuracy.

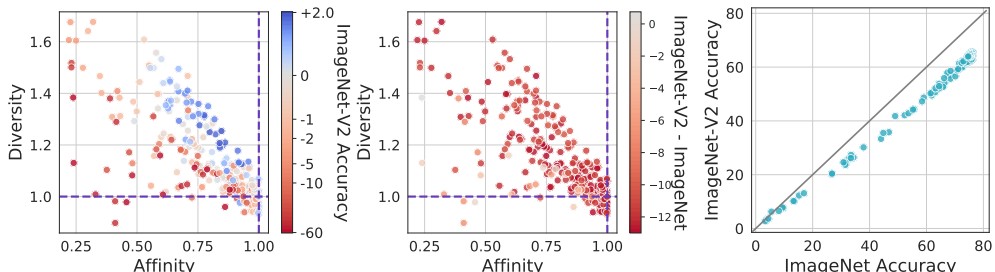

Figure 14: (Left) Augmented model performance on ImageNet-V2 relative to a clean baseline (acc 63.7). We see that robust accuracy is higher for models trained with more diverse higher affinity augmentation policies. (Middle) Gap in performance between ImageNet-V2 and ImageNet accuracy. The gap diminishes for models trained with high diversity, high affinity augmentations. (Right) ImageNet-V2 accuracy versus ImageNet accuracy.

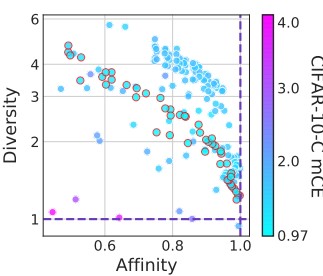

Figure 15: Mean corruption error on CIFAR-10-C (lower is better): Augmentations with high Affinity and high Diversity generally lead to lower mean corruption scores, and thus better robustness performance. Patch Gaussian augmentations (red circles), however, outperform this trend, potentially due to the similarity between some of the corruption transformations and Gaussian noise.

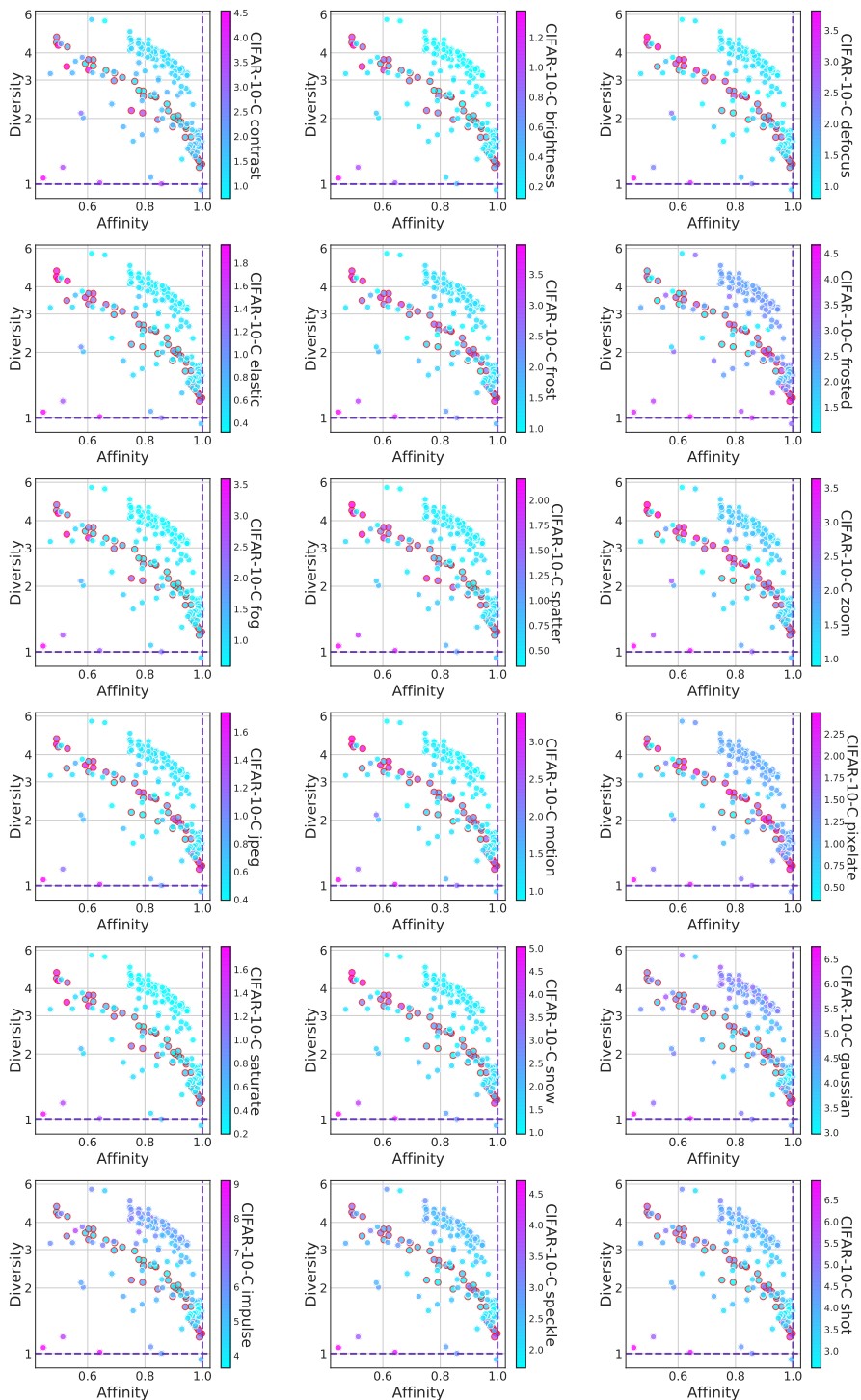

Figure 16: Robustness performance on CIFAR-10-C: Figures show corruption error for 18 different corruption types averaged over five corruption strengths and compared to the baseline model, AlexNet, used by Hendrycks & Dietterich (2019). Robust performance for some corruptions, such as brightness, is maximal for models trained with high Affinity high Diversity augmentations. For other corruptions, such Gaussian noise or speckle, we do not see this. Red circles indicate models trained with patch Gaussian augmentation which are particularly robust to paired families of augmentations.

