# OpenReview forum: "Tradeoffs in Data Augmentation: An Empirical Study"
_ICLR.cc/2021/Conference — ICLR 2021 Poster_

### Official Review · AnonReviewer4 · 2020-10-22
**Relevant topic, interesting ideas; proposal and methods can be polished**

**Rating:** 5
**Confidence:** 5

**Review:**

## Edit after authors' responses

I have upgraded my score (from 4 to 5) based on the clarifications provided by the authors and the updated manuscript. Please see the details in my extended comments: https://openreview.net/forum?id=ZcKPWuhG6wy&noteId=V7Wy0Mpsz7Q

## Summary of the paper

This paper proposes two metrics, affinity and diversity, for assessing the value and contribution of data augmentation strategies (single transformation or combinations of them). Given a model trained on a clean data set (without data augmentation), the affinity of an augmentation strategy is defined as the difference between the accuracy of the model on augmented validation data, minus the accuracy on clean validation data. The diversity of an augmentation strategy is defined as the final training loss of a model trained with data augmented according to that strategy. The paper presents an empirical analysis of the affinity and diversity of a set of image augmentation strategies evaluated on a network architecture trained on CIFAR-10 and one trained on ImageNet. The main conclusion is that the contribution to a model's performance of an augmentation strategy is predicted by its joint affinity and diversity, but not separately.

## Summary of merits and concerns

### Merits

+ The paper's overarching motivation of quantifying the usefulness of data augmentation strategies is interesting and definitely important, given the renewed interest in data augmentation by the machine learning community.
+ The proposal of quantifying the data distribution shift and complexity (affinity and diversity, respectively) introduced by an augmentation strategy is reasonable, interesting and well motivated.
+ The introduction of the problem and motivation for the proposal (Introduction) is comprehensible and interesting and the review of related work is exhaustive and relevant. This part of the paper is very well written and I quite enjoyed reading it.
+ In the rest of the paper, the concepts and definitions introduced are easy to understand and the methods employed for empirically assessing the affinity and diversity are generally clear.

### Concerns

- I see some issues in the specific definition of affinity and diversity. Summarised (extended below), first, while the affinity can be easily computed for any augmentation strategy and pre-trained model, the diversity is computationally costly as it requires re-training the model;  second, in their current definition, the dependence on the specific model and data set makes the metrics hard to compare across models and data sets; third, affinity and diversity are defined in very different ways, one in terms of the accuracy of a pre-trained model, the other in terms of the training loss.
- Although generally clear, the methodology employed falls short at demonstrating the contributions stated in the introduction and portraying a complete picture of how affinity and diversity can be used to assess the value of data augmentation strategies. We do gain some insights, but I have some concerns about the methodology and some important questions remain open.
- One motivation for the introduction of metrics to quantify the merit and mechanisms of data augmentation strategies is that the reasons why cutout, SpecAugment and mixup work so well are not well understood. Another one is that (so-called) automatic data augmentation strategies, such as AutoAugment, are hugely computationally expensive. However, the paper does not really discuss how affinity and diversity explain the mechanisms of these augmentation strategies and how they can be used to discover new strategies more efficiently.
- The presentation of the results, especially in the figures, can be improved, in my opinion, and hinders the clarity of the paper.

## Evaluation and justification

While acknowledging the merits and contributions of the paper, my concerns outweigh the positive aspects of the paper and hence my recommendation of rejection. I will discuss next in more depth these concerns in order to better justify my recommendation and with the intention to provide constructive feedback for potential subsequent work on the paper.

### Definitions of affinity and diversity

The definitions of affinity and diversity proposed in this paper are intuitive, easy to understand and reasonable. Furthermore, as stated above, I agree that quantifying the concepts and intuitions behind these metrics is an important contribution. In fact, the main result that the value of an augmentation strategy depends on both its affinity and diversity matches my (and the author's) intuitions and expectations. However, I have several concerns about the specific way these quantities are defined and computed in the paper.

First of all, since one of the motivations for proposing such metrics is to more efficiently discover new augmentation strategies and assess the existing ones, I think that an important feature of the metrics should be the efficiency and ease of computation. However, computing the diversity of an augmentation strategy for a model requires training the model end-to-end. This is not the case of the affinity, which can be computed for any pre-trained model, and I think a useful definition of the diversity should achieve the same goal.

Second, in the way affinity and diversity are defined in the paper, it is hard to compare augmentation strategies across models and data sets, even though they are identical, that is an image rotation is applied in the same way on CIFAR-10 and on ImageNet; on ResNet and on DenseNet. Thus, it would be desirable if comparisons across models and data sets would be possible. This mismatch is in fact reflected in the presentation of the results, hindering the clarity. For example, in Figures 3b and 3c, the colour codes represent very different things as the range varies in one case from -50 to +7 and in the other case from -70 to +0.6. Green dots in Figure 3b (CIFAR-10) represent augmentation strategies that improve the accuracy with respect to the baseline, while green dots in Figure 3c (ImageNet) correspond to strategies that hinder the performance. Furthermore, the range of affinity and diversity also differs greatly between the two plots. This has to do in part with a likely suboptimal way of presenting the results (more about this below), but also with the fact that the affinity and diversity in ways that are not comparable across models and data sets.

One reason why the metrics are not comparable is that they are defined in absolute terms, without any normalisation that cancels the dependence on specific aspects of the model and data set, such as the loss, the number of classes (which determines the loss and accuracy), etc. Moreover, the authors chose to define the affinity and diversity in terms of the (top-1) accuracy and the (cross-entropy) loss, respectively. However, other researchers that may wish to use these metrics in the future might prefer to assess their models using an alternative metric, such as the top-5 accuracy, commonly used for ImageNet, or trained their models with a different loss. Such choices would also affect the interpretation of affinity and diversity and complicate even further the comparisons.

A suggestion would be to define the metrics in relative terms instead. Without claiming that these suggestions would be optimal, the affinity could be computed, for instance, as the accuracy on the augmented data divided by the baseline accuracy on the clean data (and optionally multiplied by 100 to turn it into a percentage): $\mathcal{T} = A(m, D') / A(m, D) * 100$. This would represent the fraction of the percentage of the accuracy obtained by testing on augmented images. These would reduce the dependence on the specific metric (accuracy) and on the characteristics of the model and data set. This has been used for instance in [[1]](#references) to also compared the contribution to performance of models trained with different data augmentation strategies.

Third, affinity and diversity are defined in very different ways. Intuitively, the concepts that these quantities aim to represent are both related to the data distribution: affinity is related to the shift in the data distribution introduced by an augmentation strategy; diversity is related to the complexity in the distribution introduced by the augmentation. However, the former is defined in terms of the accuracy of a pre-trained model with respect to a baseline, the other in terms of the absolute training loss achieved by training with the augmentation. These are very different, unrelated quantities. Again, without claiming that the following suggestion is optimal, one idea would be to define the diversity as measure of spread (standard deviation, variance). Have the authors considered defining the diversity along the lines of the variance of the affinity or of a related quantity? This would quantify the diversity of an augmentation strategy and at the same time remove the need to train a model end-to-end, as discussed in the first point.

The authors briefly discuss entropy as an alternative measure of diversity, despite the problem of computing it for continuously-varying transformations. This is also an interesting direction which could be worth exploring.

### Results do not demonstrate all contributions

The authors list four main contributions of their paper in the introduction. The first one is the introduction of affinity and diversity as "interpretable, easy-to-compute metrics for parametrizing augmentation performance". This is satisfied, although I have discussed above some concerns about the definitions.

The second contribution is that "performance is dependent on both metrics". This is indeed reflected by the results in Figures 3b and 3c. However, I should also note in this regard that despite the large number of augmentation strategies analysed, only two network architectures, each trained on one data set, are included in the experimental setup, and some differences between the two plots could already be discussed. The claim that the performance gain introduced by an augmentation strategy increases when both the affinity and the diversity increase is well supported by the results in Figure 3b (WRN on CIFAR-10), but this is not so clear in Figure 3c (ResNet on ImageNet). As a matter of fact, the relative test accuracy in Figure 3c seems to be higher (more yellow) as diversity decreases, rather than increases, and affinity increases. It would be desirable to obtain similar results on other architectures and data sets in order to gain more evidence to support this claim.

Related to this point, I would like to note that having two sets of results (WRN on CIFAR-10 and ResNet on ImageNet), presented in Figure 3, the authors select one of them (WRN on CIFAR-10, Figure 3b), the one, out of two, that best supports the claim, for Figure 1 on the first page of the paper. This clearly introduces a selection bias that distorts the actual data (i.e. why not showing in Figure 1 the results on ImageNet?). Moreover, the range of the axes differs between Figure 1 and Figure 3b, and there is even a third version in Figure 7, at the supplementary material. I would appreciate it if the authors can comment on/clarify this.

The third contribution claims to "connect augmentation to other familiar forms of regularization". This point is addressed in Section 4.2, where the authors evaluate the performance of the models trained with data augmentation after turning off the augmentation partway through training. Although the phenomenon that performance sharply improves after turning off regularisers partway is interesting, this has been observed in previous works (reviewed by the authors) and the analysis in this paper, through the lens of affinity and diversity, does not provide new, significant insights, in my opinion. Further, taking into account the claim in the list of contributions, the analysis offers little insight about the connection of data augmentation with other forms of regularisation, beyond noting that the "slingshot effect" occurs also with data augmentation, which had been observed before.

I would like to draw the attention to certain aspects of the methodology in this section that might distort the conclusions. For example, the authors conclude from Figure 5b that "For some poor-performing augmentations, [switching off the augmentation] can actually bring the test accuracy above the baseline". However, I would like to note that the authors report that in order to obtain these results, they tested multiple switch-off points and select the one that yields the best accuracy. This introduces a clear bias in Figure 5b towards best-case scenarios, which might give the impression that the switch-off lift is a general effect or, in the best case, the magnitude of the effect will be magnified. In order to gain a more accurate picture of the phenomenon, all the available data should be considered and ideally a statistical analysis should be carried out. Another source of bias in the visualisation of the results is that in Figure 5c, "Where Switch-off Lift is negative, it is mapped to 0 on the color scale".

On the other hand, the switch-off lift effect could be explained in simpler terms, at least partially. For example, the authors write "Bad augmentations can become helpful if switched off". First of all, this could occur by pure chance and be reflected in the reported results as an artifact of the selection bias pointed out in the previous paragraph. Second, we should simply think that it is expected that turning off a bad augmentation should improve the accuracy. As a matter of fact, the augmentation strategies used to illustrate this effect are `FlipUD(100%)` (I will assume this means vertical flip) and `Rotate(fixed, 20deg,100%)`. In both cases, with the augmentation on, the model does not see the original images, so an improvement is expected if suddenly the model does see them. If the final accuracy is actually above the baseline should be analysed through a statistical analysis, rather than focusing on the best case, which might be due to pure chance. Moreover, it is also worth questioning the accuracy on clean images is actually a good baseline, since this model sees fewer different images.

Finally, the authors claim that "Switch-off Lift varies with Affinity and Diversity", from the results in Figure 5c. However, from this figure we observe that mainly, it varies with affinity. This makes sense: very low affinity is indicative of unrealistic or at least odd augmentations, which are detrimental if performed during the whole training procedure. If turned off, the model has time to fine tune on the actual images.

The temporal dynamics of training neural networks and its relation to regularisation is an interesting, active topic of research, but due to its complexity it should be analysed very rigorously in order to minimise the risk of leading ourselves astray.

The fourth contribution claims that affinity and diversity informs that "performance is only improved when a transform increases the total number of unique training examples". This aspect is addressed in Section 4.3. The authors "seek to discriminate this increase in effective dataset size from other effects". Again, this is an important and interesting topic, as well as hard to analyse, but in my opinion the methods fall short at justifying the conclusions. Here, the authors simply trained models with static augmentation and found that "For almost all tested augmentations, using static augmentation yields lower test accuracy than the clean baseline", which is not surprising because the augmented images differ from the original validation/test distribution on which the model is evaluated. However, this observation does not prove that the gain provided by data augmentation, when it does improve the performance, is due to the increase in the effective training set size. To be clear, this is likely to be the case, intuitively, but should be proven differently.

### Cutout, SpecAugment, mixup are not discussed in terms of affinity and diversity

Gaining insights about the mechanisms that make some data augmentation techniques (Cutout, SpecAugment, mixup) work better than others would be an interesting contribution. In the introduction, the authors mention this as a motivation for proposing the affinity and diversity metrics. However, although (some of) these augmentation strategies are included in the experiments, there is no specific discussion about them anywhere in the paper. Having mentioned this in the introduction as a motivation for the paper, I did miss a discussion that provided new insights about how these methods work and when.

Similarly, the authors motivate the proposal of affinity and diversity as a way to better understand the mechanisms of data augmentation and discovering new techniques more efficiently. However, beyond presenting the empirical results in Section 4, the paper does not further discuss use cases of affinity and diversity to efficiently assess the value of new techniques, or analyse commonly used strategies in terms of these metrics. For example, a widely used data augmentation strategy in computer vision is the combination of horizontal flips and vertical and horizontal translations of about 10 % of the height and width. This has been found to provide large performance gains [[1, 2, 3]](#references), while additional transformations only marginally improved the accuracy. Knowing the effectiveness of this simple augmentation strategy, it would also be interesting to analyse its affinity and diversity.

### Visualisation of results

Although this aspect has not been decisive in my evaluation of the paper, I think there is room for improvement regarding the visual presentation of the results and hopefully the following feedback, from a careful read of the article, may help make the paper stronger.

- Figure 3a: it would help to more clearly specify, perhaps directly in the plots, that the top row corresponds to CIFAR-10 and the bottom row to ImageNet.
- Figures 3b and 3c: I have commented above on this figure specifically, about the possibility of changing the definitions of affinity and diversity that would improve the comparison across models and data sets and hence the interpretability of these figures. In any case, a confusing aspect of these figures is that the colour codes define different ranges of values in each plot, which have semantically very different interpretations. For instance, light green values on 3b correspond to augmentations that improve the accuracy with respect to the baseline, while the same colour on 3c correspond to augmentations that perform worse than the baseline. Given that there is a clear central point in the colour code, zero, where the semantic interpretation changes (positive vs. negative), I would strongly suggest to use a [perceptually uniform diverging colour palette](https://seaborn.pydata.org/tutorial/color_palettes.html#perceptually-uniform-divering-palettes).
- Figure 4: I would suggest to colour-code the dots to reflect the probability of rotation. It would reduce the cognitive load to interpret the figure.
- Figure 5a: the legends could be placed inside the axes to make space for larger figures
- Figure 5b: indicate what the dash line represents

## Questions

The questions I list below are mainly intended to raise awareness about relatively minor aspects of the paper that remained unclear to me while reading it, with the aim of providing constructive feedback to potentially improve the manuscript. The aspects that have been more decisive in my decision have been already commented above.

- "Random crop was also applied on all ImageNet models.": Why is not random crop considered data augmentation in this paper?
- How would the authors explain the strange variation of diversity with respect to the probability of rotation in Figure 4 centre? Is this a general behaviour in other augmentation strategies? Is the affinity as clear in other cases?
- Figure 5c: how is it possible to get an improvement (Switch-Off Lift) of 50 %, while in Figure 5b all cases show smaller variation?
- The specific version of the wide residual network reported in the paper is "WRN 28-2" However, 28-2 is not described in the original WRN paper, and is also not described in the github repository of AutoAugment. I suppose that it should instead read WRN 28-10. Assuming this is the case, I have an additional question: WRN 28-10 achieves, according to my own implementation, around 91.5 % accuracy on CIFAR-10 without any data augmentation, using the hyperparameters and regularisation of the original paper. If I interpret correctly Figure 5b, the baseline accuracy achieved by the authors is 89.7, which is significantly lower. I would appreciate it if the authors could clarify what I may have misunderstood.
- In Section 4.3, what is the goal of comparing models trained with static vs dynamic data augmentation? How would models trained with static augmentation be better or have larger diversity?
- "transforms and hyperparameters from the AutoAugment search space [...] implicitly have high Affinity": this is not what we see in the Figure 3b and Figure 7. Could the authors clarify this statement?

## Minor comments and potential typos identified

- "some have proposed that augmentation strategies are effective because they increase the diversity of images seen by the model": any reference where this claim is made?
- Make sure that "dataset"/"data set" are spelled consistency throughout the article.
- Would the authors venture any guess about the affinity and diversity of strategies in other data domains?
+ I appreciate that the authors report (Section 3) details about how the test results were computed, including the standard errors
- "static training" (Section 3): Do the authors mean "static augmentation"? Also, consider giving an example for better illustration.
- It is slightly confusing that the augmentation function is denoted by $a$ in Definition 1, which is an unusual choice for a function. Would a capital letter be a better choice, since augmentations are generally stochastic functions?
- "KL divergence of the shifted data with respect to the original data": consider specifying this mathematically too
- What is the reason for the capitalisation of _Affinity_ and _Diversity_?
- Typo: "model-dependant" (Section 3.1)
- The gap between Figure 5's caption and the paragraph seems to have been manually reduced. If this is the case, it may be against the formatting guidelines and, especially, it hinders the readability.
- Some data augmentation strategies are mentioned without previously introducing them. For example, `FlipUD` in Section 4.2

## References

[1] Hernández-García, Alex, and Peter König. "Data augmentation instead of explicit regularization." arXiv preprint arXiv:1806.03852, 2018.

[2] Goodfellow, Ian, et al. "Maxout networks." International conference on machine learning. PMLR, 2013.

[3] Springenberg, Jost Tobias, et al. "Striving for simplicity: The all convolutional net." arXiv preprint arXiv:1412.6806, 2014.

---

> ### Author Response · Authors · 2020-11-19
> **Re: Relevant topic, interesting ideas; proposal and methods can be polished (Part 1)**
>
> We thank the reviewer for taking the time to provide an exceptionally in depth review of our work. Though we do not agree with all points (see below for some push back), we do feel the reviewer has raised some important points which have helped to strengthen the manuscript.
>
> _diversity is computationally costly as it requires re-training the model_
>
> You are absolutely correct that evaluating diversity as defined in equation (2) for each augmentation policy requires training a new model and is costly. For this reason, in the paper we introduced an alternative measure of diversity, the entropy of an augmentation, this has the advantage that it can be computed without training any model, but has the disadvantage of being ill defined for continuous augmentations.
>
> To remedy this shortcoming further, we have included an alternative computationally less-expensive proxy for diversity, shown in in Figures 10 and 11. We found that the training loss at a much earlier epoch correlates well with the final training loss and so can serve as an alternative measure of diversity. This still requires training a model for each augmentation policy, but only for a small fraction of the training time. In particular the results shown in Fig 10 are for a model trained for 10 epochs, a 95% reduction in computational cost.
>
> The reviewer also proposed an alternative interesting, inexpensive candidate diversity measure:
>
> _one idea would be to define the diversity as a measure of the spread (standard deviation, variance). Have the authors considered defining the diversity along the lines of the variance of the affinity or of a related quantity?_
>
> Thank you for the suggestion, we had not considered this, and its computational efficiency would be appealing. We were not sure if the review was suggesting variance over model initializations, or over instances of the augmented dataset. We assumed the latter, but for completeness, following your suggestion, we have implemented this measure with variance computed over model reinitializations and are currently measuring variance over dataset instances (via batches). Unfortunately, the variance over initializations does not correlate well with the other diversity measures. We have updated the manuscript with the model variance measurements (supplementary section H) and will upload a revised version when the data variance is complete.
>
> _the dependence on the specific model and data set makes the metrics hard to compare across models and data sets_
>
> It is definitely true that the metrics vary between datasets. One way to address the variation of the metric ranges is to normalize as you have suggested. We are happy to do this and have discussed below.
>
> A higher level point is that variation in the relative ordering of augmentations between datasets is expected and a feature of this approach. Even within image classification It is not true that augmentation strategies are equally effective from dataset to dataset. For example, the AutoAugment policy found on SVHN did not perform well on CIFAR-10 (see AutoAugment and Unsupervised Data Augmentation), and Cutout does not perform well on ImageNet while being very effective on CIFAR-10 and SVHN (see Cutout, Patch Gaussian).  It is reassuring that this is reflected in Affinity and Diversity. For a fixed dataset, we expect the relative ordering of augmentations to be relatively stable to model details. This assumption has been used with success (e.g. see AutoAugment, Population Based Augmentation, Fast AutoAugment) to identify successful augmentation strategies on one architecture and use them on another.
>
> _One reason why the metrics are not comparable is that they are defined in absolute terms … A suggestion would be to define the metrics in relative terms instead_
>
> Thank you for the suggestion. We initially opted for basing Affinity off of absolute accuracies as many researchers have developed intuition for ImageNet and CIFAR10 raw accuracies. Nonetheless, we do appreciate the value in the suggested relative metrics! We have presented example plots in supplementary section H based off of normalized Affinity and Diversity and would love to hear from reviewers which normalization they prefer!
>
> _affinity and diversity are defined in very different ways … these are very different, unrelated quantities_
>
> Affinity is the performance (measured by accuracy) of a clean model on an augmented test set. Diversity is the performance (measured by loss) of an augmented model on the augmented train set. Though a bit less convenient, we can consider using loss, rather than accuracy to measure affinity. In this case, Affinity and Diversity are related by exchanging train and test set, and clean and augmented models.

---

> > ### Author Response · Authors · 2020-11-19
> > **Re: Relevant topic, interesting ideas; proposal and methods can be polished (Part 2)**
> >
> > _the paper does not really discuss how affinity and diversity explain the mechanisms of [SpecAugment, mixup, and AutoAugment] augmentation strategies and how they can be used to discover new strategies more efficiently._
> >
> > We believe that our analysis sheds light on the mechanisms of vision augmentations, such as mixup, Cutout, and AutoAugment in several ways. First of all, it is worth noting that many of the important methods that built on AutoAugment focused on one of our proposed metrics. For example, Fast AutoAugment and CTAugment focus on increasing the Affinity of an AutoAugment policy, whereas Adversarial AutoAugment focuses on increasing the diversity of an AutoAugment policy. Our work is the first one to propose and study these two metrics generally, which helps put previous work in context. Furthermore, our work suggests that further improvements can be expected if one tries to increase both of the metrics, instead of just one of them as was the case in previous work. Given our work, an obvious strategy to find better policies would be to combine the losses of Adversarial AutoAugment and Fast AutoAugment, which we plan to do in future work. Finally, and knowing our insights from the vision domain, we speculate that a similar mechanism might be at play in other domains, such as speech recognition, and underlie why strategies such as SpecAugment, which appear to be relatively out of distribution, nonetheless work -- we predict that they drastically increase diversity.
> >
> > _combination of horizontal flips and vertical and horizontal translations ... has been found to provide large performance gains [1, 2, 3]_
> >
> > We thank you for bringing this up. We actually have included this combination of augmentations, we call them FlipLR+Crop (because this operation on CIFAR-10 is commonly called Flips and Pad-and-Crop, which is the same operation you mention as horizontal flips and vertical and horizontal translations). It is worth noting that these two augmentations are two of the highest Affinity augmentations, which might explain why they have been so commonly used on CIFAR-10 and CIFAR-100. Below we list accuracy, Affinity, and Diversity values for the combination of these two augmentations. We have also added these references to the paper.
> >
> > Acc: 94.6%, Affinity: -2.97, Diversity: 0.105
> >
> > _The claim that the performance gain introduced by an augmentation strategy increases when both the affinity and the diversity increase is well supported by the results in Figure 3b (WRN on CIFAR-10), but this is not so clear in Figure 3c (ResNet on ImageNet)._
> >
> > Thanks for raising this point. One shortcoming of the presentation in figures 3b and 3c is that it relies on visual clarity rather than a sharp metric for the take-away message.
> >
> > To make this more precise we have now included a measure of the fraction of pairs of augmentations (a,a’) satisfying the condition that if Acc(a) > Acc(a’) then either Aff(a) > Aff(a’) or Div(a) > Div(a’). The ImageNet plot (Figure 3c) satisfies this for 97.5% of all pairs (compared with 99.1% of pairs for CIFAR-10 and 75% for an uncorrelated null model).
> >
> > _the authors select one [set of results] for Figure 1_
> >
> > We have now included both CIFAR-10 and ImageNet results in Figure 1
> >
> > _the range of the axes differs between Figure 1 and Figure 3b_
> >
> > Figure 3b includes the modern augmentation policies, RandAugment, mixup, and AutoAugment, while Figure 1 does not. We adjusted the axis to accommodate these exceptionally high Affinity/Diversity augmentations.

---

> > > ### Author Response · Authors · 2020-11-19
> > > **Re: Relevant topic, interesting ideas; proposal and methods can be polished (Part 3)**
> > >
> > > _The authors conclude from Figure 5b that "For some poor-performing augmentations, [switching off the augmentation] can actually bring the test accuracy above the baseline". However, I would like to note that the authors report that in order to obtain these results, they tested multiple switch-off points and select the one that yields the best accuracy. This introduces a clear bias._
> > >
> > > We apologize if this was not clear and believe the reviewer may be misunderstanding our claim. As described in supplementary section D, we follow the standard procedure of tuning a hyperparameter (the time at which to switch off an augmentation policy) on a held out validation set and then report the corresponding test performance. The standard error on the mean over the ten independent runs are so small as to not be visible in figure 5a. This has now been clarified in the caption to Figure 5. This clearly demonstrates that there exist cases where switching off a poor-performing augmentation policy can improve over baseline performance. To make this more clear we have also included a table in supplementary section D with the performance for standard and optimally switched augmentation strategies as well as the std error on the mean for all augmentations studied 43% show a benefit from switching at at least 2X the standard error, 47% show a change relative to standard augmentation within +/- 2X the standard error, and 10% show a decrease.
> > >
> > > We make no claim that switching of a bad augmentation is beneficial if done at an arbitrary time point during training. If published, we will release a link to the full data containing performance for these sub-optimal switching times.
> > >
> > > _we should simply think that it is expected that turning off a bad augmentation should improve the accuracy._
> > >
> > > The interesting phenomena is not that the performance improves, but that it improves over baseline performance. We found 29 examples of this and 20 where the improvement over the baseline was greater than 2X the standard error.
> > >
> > > _If the final accuracy is actually above the baseline should be analysed through a statistical analysis, rather than focusing on the best case, which might be due to pure chance._
> > >
> > > As discussed above, selection was performed on a held out validation set and performance is statistically significantly above the baseline, inconsistent with arising from statistical fluctuations.
> > >
> > > _Visualisation of results: hopefully the following feedback, from a careful read of the article, may help make the paper stronger._
> > >
> > > Thank you for the feedback. We have incorporated all of these suggestions into the revised manuscript and agree it makes the paper stronger. In particular, Figures 1, 3, and 4 now use a color scheme with a consistent zero. As mentioned above, for the rescaled Affinity and Diversity definitions we have provided sample figures in supplementary section H. We have also provided examples with different color schemes (Fig 18) and would appreciate any feedback from the reviewers on preferences.
> > >
> > > ### Questions
> > > _"Random crop was also applied on all ImageNet models.": Why is not random crop considered data augmentation in this paper?_
> > >
> > > In common ImageNet infrastructures, random crop is integrated into the data loading pipeline, which makes it hard to disentangle it as a separate augmentation. Furthermore, models trained on ImageNet without random crop perform very poorly. In order to analyze the effect of augmentations against a more realistic baseline and due to the infrastructure constraints, we chose to keep random crop preprocessing.
> > >
> > > _How would the authors explain the strange variation of diversity with respect to the probability of rotation in Figure 4 centre? Is this a general behaviour in other augmentation strategies? Is the affinity as clear in other cases?_
> > >
> > > We assume the reviewer is asking about the lack of monotonicity of Diversity with respect to augmentation probability. We do not have a deep understanding of this behavior, nor do we have an argument that training loss should be strictly monotonic in augmentation probability. Some degree of non-monotonicity appears common across augmentations.
> > >
> > > For affinity there is a simple argument that ensures monotonicity with probability -- the affinity of a policy applying an augmentation with probability p takes the form A = p A_on + (1-p) A_off = p A_on, where A_on is the affinity with the augmentation applied with probability 1 and A_off is the affinity with no augmentation (which is equal to zero). This ensures the monotonicity for these single probability augmentations.

---

> > > > ### Author Response · Authors · 2020-11-19
> > > > **Re: Relevant topic, interesting ideas; proposal and methods can be polished (Part 4)**
> > > >
> > > > _Figure 5c: how is it possible to get an improvement (Switch-Off Lift) of 50 %, while in Figure 5b all cases show smaller variation?_
> > > >
> > > > The x-axis in Figure 5b uses a symlog scaling, with linear threshold -1 and +1 (now indicated in the Figure with grey dashed lines. The left-most point represents a Switching Lift of +50. We realize that in the original Figure, the x-axis was missing a label for the left-most point (Aug On accuracy 39%), that is now indicated in the figure as well. Thanks for catching this!
> > > >
> > > > _The network reported in the paper is WRN 28-2 … should [it] instead read WRN 28-10. _
> > > >
> > > > The model we use is WRN 28-2. In the original wide resnet paper, the authors introduce the notation WRN-n-k to denote a residual network that has a total number of convolutional layers n and a widening factor k.
> > > >
> > > > _In Section 4.3, what is the goal of comparing models trained with static vs dynamic data augmentation? How would models trained with static augmentation be better or have larger diversity?_
> > > >
> > > > The goal of training models with static augmentation is to see whether broadening the distribution off the strict training manifold without introducing additional samples provides any benefit (relative to un-augmented data). The goal of comparing static and dynamic augmentations to each other is not because we expect static augmentats to be better or more diverse, but to quantify the boost in diversity that the stochastic policy has relative to the fixed policy.
> > > >
> > > > _"transforms and hyperparameters from the AutoAugment search space [...] implicitly have high Affinity": this is not what we see in the Figure 3b and Figure 7. Could the authors clarify this statement?_
> > > >
> > > > We thank the referee for bringing this up, we changed this in the text. We agree that the AutoAugment policy is not necessarily high Affinity compared to all the ops we considered. We wanted to make the point that the while Adversarial AutoAugment method aims to increase diversity, this aim alone cannot lead to effective augmentation strategies. For example, setting the pixels of all training images to black would maximize the training loss (diversity), but would lead to terrible generalization. However, Adversarial AutoAugment tries to maximize diversity while restricted to AutoAugment ops and hyperparameters, which are higher affinity than a random transformation one can apply to an image. For example, the range of rotations in AutoAugment is set to be -30 to 30 degrees, which is higher Affinity than a different range such as 30 to 60 degrees or -60 to 60 degrees. However, the referee is correct that AutoAugment ops are not necessarily as high affinity as the other high affinity operations such as horizontal flips.
> > > >
> > > > ### Minor comments and potential typos identified
> > > >
> > > > _"some have proposed that augmentation strategies are effective because they increase the diversity of images seen by the model": any reference where this claim is made?_
> > > >
> > > > One classic reference is Simard et. al. 2003
> > > >
> > > > _Would the authors venture any guess about the affinity and diversity of strategies in other data domains?_
> > > >
> > > > At the level of speculation, we expect a similar tradeoff in other domains. SpecAugment is one concrete example in speech recognition which we expect to be the analogue of a high diversity lower affinity augmentation -- the authors note the increase in training loss and out of distribution character. In fact, one of our long term motivations for attempting to find well defined metrics for augmentation policies is to hopefully port the success of augmentations in image understanding to domains which have yet to realize such big gains. This is an active direction of future work.
> > > >
> > > > _It is slightly confusing that the augmentation function is denoted by  a  in Definition 1, which is an unusual choice for a function. Would a capital letter be a better choice, since augmentations are generally stochastic functions?_
> > > >
> > > > We wanted to avoid the confusion created by using capital A for both Affinity and augmentation.
> > > >
> > > > _What is the reason for the capitalisation of Affinity and Diversity?_
> > > >
> > > > We wanted to introduce precise terms with sharp definitions, and did not want the terms to be confused with their meaning in colloquial language.
> > > >
> > > > ### Typos and other minor comments
> > > >
> > > > Thanks for catching these, we have implemented them in the revised text.
> > > >
> > > > Thank you again for your feedback! Are there any other concerns we didn’t address?

---

> > ### Comment · AnonReviewer4 · 2020-11-23
> > **Upgraded score, though main concerns remain**
> >
> > I sincerely appreciate the effort and time the authors have employed in addressing most of the concerns of my (rather long) review, as well as updating the manuscript. I am glad if the authors agreed with some of my suggestions. In particular, I positively value the improvements regarding clarity, especially the figures. For these reasons, I have upgraded my evaluation of the submission in one point. Unfortunately, I still assess the manuscript marginally below the acceptance threshold, since my concerns regarding the definitions of affinity and diversity remain, as well as how these metrics shed light to our understanding of data augmentation. I think that my views on these issues was largely explained in my original review, but I will summarise it below, taking into account the responses of the authors.
> >
> > First of all, as stated in my original review, I see a lot of potential in the proposal of such metrics to quantify the contribution of data augmentation on generalisation. I think that the notions of affinity and diversity, as intuitively depicted in Figure 1c (of the updated submission), are meaningful and  potentially useful, at least to my intuition. However, in particular the definition of diversity as the final loss of a model, diverges from the notion reflected in Figure 1c. Furthermore, it is computational expensive (although the authors have studied a less costly alternative; see below) and of different nature than the affinity.
> >
> > In their response, the authors have mentioned some new results from studying "early diversity", that is the loss computed after a few epochs, which is correlated with the final loss. This is interesting, but it introduces additional problems: Is the difference with respect to the final loss negligible? How many epochs are acceptable for computing the early diversity? I humbly think that an alternative may be to re-defining the diversity, more aligned with the affinity and especially with the notion from Figure 1c. I am willing to elaborate on the suggestions made in my original review, in case the authors find them useful, and discuss these ideas in a separate thread.
> >
> > Regarding my concern that the claim that "performance is dependent on both metrics", the authors have introduced a new result, the count of pairs of augmentations for which "if Acc(a) > Acc(a') then either Aff(a) > Aff(a') or Div(a) > Div(a')". However, if the claim is that performance depends on both metrics, should the condition not be Aff(a) > Aff(a') AND Div(a) > Div(a'), that is AND, not OR?
> >
> > Another concern I raised was that the phenomenon of improved performance after turning off data augmentation (Section 4.2) is only vaguely connected to affinity and diversity. In their response, the authors have resolved some of my questions about the methodology in this section, but the main concern remains. Furthermore, the authors have not addressed my concerns regarding the fourth contribution, that the analysis in Section 4.3 falls short at answering the question of whether the gain provided by data augmentation is due to an increase in the effective training size, or by what extent.
> >
> > The authors have provided some insights about how affinity and diversity may help us understand some augmentation strategies such as mixup, SpecAugment and AutoAugment. I appreciate this, but I think it would be interesting to discuss this in the paper, as  it is one of the contributions claimed in the introduction.
> >
> > Finally, some unrelated comments:
> >
> > * I agree that "random crop is integrated into the data loading pipeline, which makes it hard to disentangle it as a separate augmentation". And the fact that training without random crop yields bad results speaks for the importance of data augmentation, diversity, in this case. However, not considering leaving random crop as part of the training process of the baseline models, without considering it data augmentation, may strongly distort the metrics. Actually, the little improvement of in test accuracy by augmentations on ImageNet (Figure 3c) could be explained by the fact that the random crops are providing most of the data augmentation-related gains.
> > * I personally prefer the diverging colour scheme of Figure 18 top.
> > * In Figure 1, 1b has a grid, 1a not.
> > * Absolute vs relative: "many researchers have developed intuition for ImageNet and CIFAR10 raw accuracies": that's right, but it doesn't mean it's the best approach, is it?

---

> > > ### Author Response · Authors · 2020-11-24
> > > **re: Upgraded score, though main concerns remain**
> > >
> > > Thank you for taking the time to read the response and for the additional comments and clarification (as well as the score raise)! We have clarified and responded to specific pieces below.
> > >
> > > _I humbly think that an alternative may be to re-defining the diversity ... I am willing to elaborate on the suggestions._
> > >
> > > We would be excited to hear the full suggestion in another thread! We have now included a preliminary analysis showing the correlation between Diversity defined as the loss at the end of training, and the variance of affinity over the dataset for some CIFAR10 augmentations. We find poor correlation between the two in this case (new Figure 20).
> > >
> > > _the authors have introduced a new result, the count of pairs of augmentations for which "if Acc(a) > Acc(a') then either Aff(a) > Aff(a') or Div(a) > Div(a')". However, if the claim is that performance depends on both metrics, should the condition not be Aff(a) > Aff(a') AND Div(a) > Div(a'), that is AND, not OR?_
> > >
> > > Our claim was indeed that performance depends both on Affinity and Diversity. It was not, however, intended to imply that if one augmentation was better than another it needed to have both higher affinity and higher diversity. Rather improving affinity or improving diversity both help performance, and the inequality is intended to capture this. Perhaps the message is more clear when phrased using the contrapositive: what fraction of pairs of augmentations violate this picture -- i.e. what fraction have Acc(a) >  Acc(a’) but where a has both lower Affinity and lower Diversity than a’. This is 1 - the numbers quoted in the text. Ie only 2.5% of pairs for ImageNet and .9% of pairs for CIFAR10 violate this.
> > >
> > > _The authors have provided some insights about how affinity and diversity may help us understand some augmentation strategies such as mixup, SpecAugment and AutoAugment. … it would be interesting to discuss this in the paper._
> > >
> > > Apologies for missing this! We had intended to, and now have, expanded the analysis in the discussion.
> > >
> > > _unrelated comments_
> > >
> > > For all of the plotting suggestions -- color scheme, relative Affinity and Diversity, grid consistency. Unless we hear strong opposing viewpoints from other reviewers, if accepted, we are happy to make these changes in the camera ready version.

---

> > > > ### Comment · AnonReviewer4 · 2020-11-24
> > > > **if Acc(a) > Acc(a') then Aff(a) > Aff(a') ? Div(a) > Div(a')**
> > > >
> > > > Thanks for the new clarifications.
> > > >
> > > > Regarding the boolean relationship between [Aff(a) > Aff(a')] and [Div(a) > Div(a')] if Acc(a) > Acc(a'), contribution 2 of the paper is the following: "We find that performance is dependent on _both_ metrics. In the Affinity-Diversity plane, the best augmentation strategies jointly optimize the two (see Fig 1)". From _both_ and _jointly_ this I understand that the two metrics should generally correlate with better performance, and for a majority of pairs, a and a', if Acc(a) > Acc(a') then _both_ Aff(a) > Aff(a') AND Div(a) > Div(a'). Certainly, these are noisy measures and I would not expect a value close to 100 %, but definitely a majority. From Figure 3b, it seems that this would be the case for CIFAR-10, supporting the claim, but by a much lesser extent for ImageNet (Figure 3c). This is precisely the concern raised in my original review.
> > > >
> > > > I concede that the claim holds true if for the same value of diversity, higher affinity leads to better accuracy (for the same value of affinity, higher diversity leads to better accuracy). This would make the conditional statement more precise with 'greater than or equal' operators. Still, that would only capture a few cases of continuously varying metrics. Perhaps a more informative summary statistic should take into account differences, not boolean relationships.

---

### Official Review · AnonReviewer2 · 2020-10-24
**This paper is interesting and can be improved.**

**Rating:** 6
**Confidence:** 4

**Review:**

# Summary

This paper analyzes data augmentation for image classification using two measures: Affinity and Diversity. These measures depend on both training data and model and are easy to compute. The authors show that the performance of image classifiers depends on both of these measures through extensive experiments.

# Strengths

* The introduced measures consider both data and models, which are inseparable in modern deep learning.
* The measures can explain why some data augmentation methods work and others are not, intuitively.
* Affinity is easy to compute. To obtain Affinity, one needs a model trained on clean data and uses validation sets with a given augmentation. One can reuse the trained model to measure other augmentations.
* The experiments are extensive.

# Weaknesses

* Diversity requires to train a CNN model with each augmentation configuration. This requirement restricts the crucial application of the paired measures to find better augmentation configurations given a dataset and a model under limited computational cost.
* The slingshot effect in section 4.2 is interesting. But Fig 5 (c) shows a weak connection between the slingshot effect and the proposed measures. Instead, I think the results suggest the existence of other factors. Explaining this effect by the proposed measures only would be difficult, and I recommend removing this subsection.
* Section 4.3 shows dynamic augmentations increase the effective training data size (compared to static augmentations), and thus the performance improves. I think this is well-known, and that's why we call data augmentation "data augmentation."
* Experiments are conducted only on CIFAR-10 and ImageNet. I know these datasets are the de-facto standard, but datasets from other domains are preferable to be included to support the claims.

# Rating

5. This paper includes interesting and useful insights, but its novelty is limited. Besides, its applicability is also restricted because of the computational intensity of the Diversity measure.

# Feedbacks

* Titles for the color bars in Fig 2 are missing.
* It took moments to understand T and B in Figure 3 (a) are top and bottom.
* The margin between the caption and the main text on page 7 is too small, which confused me during reviewing.
* Reference information is old. Lim et al. 2019 is accepted at NeurIPS, and Hataya et al. 2020 is accepted at ECCV.

---

> ### Author Response · Authors · 2020-11-19
> **Re: This paper is interesting and can be improved.**
>
> We thank the reviewer for taking the time to read our work and for providing constructive comments. We believe the feedback regarding computational cost has especially improved the manuscript!
>
> _Diversity ... restricts the crucial application of the paired measures to find better augmentation configurations ... under limited computational cost._
>
> You are absolutely correct that evaluating diversity as defined in equation (2) for each augmentation policy requires training a new model and is costly. For this reason, in the paper we introduced an alternative measure of diversity, the entropy of an augmentation, this has the advantage that it can be computed without training any model, but has the disadvantage of being ill defined for continuous augmentations.
>
> To remedy this shortcoming further, we have included an alternative computationally less-expensive proxy for diversity, shown in in Figures 10 and 11. We found that the training loss at a much earlier epoch correlates well with the final training loss and so can serve as an alternative measure of diversity. This still requires training a model for each augmentation policy, but only for a small fraction of the training time. In particular the results shown in Fig 10 are for a model trained for 10 epochs, a 95% reduction in computational cost.
>
> It is also worth noting that in many cases good augmentation strategies generalize across models. This reusability can help alleviate the initial cost of finding a successful strategy.
>
> _The slingshot effect in section 4.2 is interesting. But Fig 5 (c) shows a weak connection between the slingshot effect and the proposed measures._
>
> Thanks for the comment, though to our eye the slingshot effect does seem to be least for the high affinity high diversity augmentations, this can easily be explained by the lift correlating with accuracy itself, thus we agree that figure 5c does not add much additional insight. We have thus removed figure 5c and changed the tone of section 4.2 to emphasize the main points i) that the switching lift is a phenomena which is common to augmentations and other more classical forms of regularization ii) That in some cases naively deteremtal augmentations can improve over baseline performance when switched off.
>
> _Section 4.3 shows dynamic augmentations increase the effective training data size ... this is well-known_
>
> The intuition that data augmentation increases the effective dataset size is indeed well known, however it has also been suggested that data augmentation perhaps plays an additional beneficial regularizing role, separate from the increase in dataset size. The aim of this section is to test a strong version of this hypothesis concretely. We find a negative result -- no evidence for this independent benefit of augmentation, but still feel it is a worthwhile addition to the literature.
>
> _Experiments are conducted only on CIFAR-10 and ImageNet. I know these datasets are the de-facto standard, but datasets from other domains are preferable to be included to support the claims._
>
> We agree that investigating the utility of Affinity and Diversity for augmentations in other domains is an exciting direction! Unfortunately a thorough investigation of other domains is beyond our scope here.
>
> Nonetheless, to address this point, we are performing additional experiments on street view house numbers to broaden the datasets used and will update the manuscript shortly. If the reviewer has a particular additional domain request, we would be happy to include experiments if it is within our infrastructural capabilities and time constraints.
>
> _Feedbacks_
>
> Thank you for the feedback. We have corrected all of the points mentioned here in our updated manuscript.
>
> Thank you again for your constructive feedback! Are there any other concerns we didn’t address? If we have addressed your concerns, we humbly ask if you would kindly consider supporting accepting the paper by increasing the review score accordingly

---

### Official Review · AnonReviewer1 · 2020-10-27
**The paper introduces novel concepts to make the effect of data augmentation predictable.**

**Rating:** 8
**Confidence:** 4

**Review:**

##########################################################################

Summary:

This paper studies the problem of data augmentation that obtains new training examples by modifying existing ones. Data augmentation is popular in machine learning and artificial intelligence since it enhances the number of training examples. However, its effect on model performance remains unknown in practice. An augmentation operator (e.g. image rotation) can be either helpful or harmful. This paper introduces two novel metrics, named affinity and diversity, to quantify the effect of any given augmentation operator. The authors find that an operator with high affinity score and high diversity score leads to the best performance improvement.


##########################################################################

Reasons for score:

Overall, I like the idea of this paper about making the effects of augmentation operators tractable. The proposed affinity and diversity scores are great indicators to evaluate the usefulness of an arbitrary operator. The finding that higher scores are better is insightful. With that, practitioners are able to inspect a large number of operators and select the most effective ones.

However, the computation of affinity and diversity scores may be very expensive. The runtime could be more than that runs an operator directly to get the performance improvement. It will be good if the authors can compare the efficiency and/or share some comments in the rebuttal.

##########################################################################

Pros:

1. The paper proposes a novel idea of quantifying the effect of data augmentation. Specifically, the idea introduces two metrics, affinity and diversity scores, to evaluate any given augmentation operator in its effect on model performance. The experiments showed that higher the scores, better the performance improvement. The finding is novel that should be the first time in the field.

2. The paper justifies that either affinity or diversity alone does not predict model performance. They together can make the prediction deterministic. The finding suggests that it is nontrivial to quantify the effect of augmentation operators. The paper nicely leverages heat maps as shown in Figure 3(b) and 3(c) to reveal the finding.


##########################################################################

Cons:

I have a concern about computation efficiency. It seems it takes significant time to compute the proposed affinity and diversity scores. For a given operator, Definition 1 regarding affinity requires running the model once. Similarly, Definition 2 regarding diversity requires running the model another time. So it needs to run the model twice to get affinity and diversity scores. Empirically, we can just run the model once to calculate the actual model improvement of the operator. If so, it is inefficient to use affinity and diversity scores for the purpose of predicting model performance.


##########################################################################

Questions during rebuttal:

It will be good if the authors can add experiments or discussions related to efficiency in runtime. Or the authors can extend the use cases beyond model performance prediction.

---

> ### Author Response · Authors · 2020-11-19
> **Re: The paper introduces novel concepts to make the effect of data augmentation predictable.**
>
> We thank the reviewer for taking the time to read our work and for providing constructive comments. We believe the feedback regarding computational cost has especially improved the manuscript!
>
> _It seems it takes significant time to compute the proposed affinity and diversity scores. … It will be good if the authors can add experiments or discussions related to efficiency in runtime._
>
> This is a good question and related to points raised by other reviewers.
>
> We have included an expanded discussion of the computational cost of these metrics in section 3.2 and appendix E. Affinity is relatively inexpensive to compute. Affinity requires training a model on clean data a single time. Diversity as defined in equation (2) is indeed expensive to compute, it requires training an independent model on each potential augmentation. For this reason in Appendix E we proposed an alternative proxy for diversity, the entropy of an augmentation. This correlates well with Diversity, but has the advantage of not requiring any additional training. The downside is that entropy is ill defined for continuous augmentations.
>
> To remedy this shortcoming further, we have included an alternative computationally less-expensive proxy for diversity, shown in in Figures 10 and 11. We found that the training loss at a much earlier epoch correlates well with the final training loss and so can serve as an alternative measure of diversity. This still requires training a model for each augmentation policy, but only for a small fraction of the training time. In particular the results shown in Fig 10 are for a model trained for 10 epochs, a 95% reduction in computational cost.
>
> It is also worth noting that in many cases good augmentation strategies generalize across models. This reusability can help alleviate the initial cost of finding a successful strategy.
>
> _Or the authors can extend the use cases beyond model performance prediction._
>
> Aside from performance prediction, we can envision that this analysis may be useful more directly when searching for better augmentation policies, especially in new domains. Although this is outside the scope of this work, we are interested in exploring this question. An early finding by Zhang et al. (2019) shows that when an augmentation strategy is guided by an increase in Diversity, performance is greatly improved. We are also optimistic that these metrics will serve as interesting ways to characterize distribution shift of the underlying input data independent of direct concerns of model performance.
>
> Thank you again for your feedback! Are there any other concerns we didn’t address?

---

### Official Review · AnonReviewer3 · 2020-10-28
**Official Blind Review #3**

**Rating:** 6
**Confidence:** 4

**Review:**

This paper empirically investigates two crucial factors: affinity and diversity in useful data augmentation strategies. Through extensive experiments on existing image augmentation methods, it demonstrates that a good augmentation practice should bring high affinity and diversity for validation and training data. Specifically, it uses the accuracy gap between augmented and clean validation data to measure affinity. The diversity is measured by final training loss with the augmentations used in training.

Pros
1. The paper is well-written and easy-to-follow.
2. The proposed two measurements for affinity and diversity are easy to compute and observe. They are both model-based, making the measures more adaptive to model biases.
3. The experiments test extensive augmentation methods to support the affinity and diversity claims.

Cons:
1. Adversarial examples are perceptually similar to the original data but can result in low validation accuracy. The proposed affinity guide may not help guide robust model training.
2. In Figure 3, the highlights of the three augmentation methods, i.e., Mixup, AutoAugment, and RandAugment, are not difficult to visualize.
3. In Figure 4, one diversity value (x-axis) seems to correspond to multiple test accuracy measurements, which looks confusing.
4. Although both affinity and diversity are essential to measuring augmentation quality, it is unclear how to use them to guide training. It seems that both positive and negative affinity values may help, according to Figure 3. Moreover, higher training loss indicates higher diversity. But high training loss may also mean a model still does not converge. It may be difficult for a new task with little experience to use the two metrics to select useful augmentation strategies.

Summary

The paper investigates two important factors (affinity and diversity) in measuring the effectiveness of data augmentation. It also provides two simple metrics to measure them. My main concern is how to use them in practice to guide training. Moreover, the affinity measure seems not helpful for robust training since adversarial examples can easily fail a model, resulting in low validation accuracy. But using them in training can help advance model robustness. The paper would be better if it can address the two concerns. Overall, I recommend it for acceptance if it can include discussions on these two concerns.

Post-rebuttal updates

Thank the authors for the efforts in answering the questions. The responses have addressed my concerns about the robust training and using training loss to measure diversity. Exploring the data augmentation for robust training will be interesting. Besides,  training loss is usually sensitive to some other hyper-parameters such as optimizer and learning rate. So, investigating the robustness of this metric is also meaningful.

Moreover, there is still a gap from applying the proposed metrics to training guidance. The current research mainly shows there are relations between the two metrics and data augmentation effectiveness.  How to merge them into one easily observable is still missing. Therefore, I keep my original score.

---

> ### Author Response · Authors · 2020-11-19
> **Re: Official Blind Review #3**
>
> We thank the reviewer for taking the time to read our work and for providing constructive feedback! We think the questions and suggestions about robustness in particular have strengthened the work.
>
> _The proposed affinity guide may not help guide robust model training._
>
> This is a valid point. In the paper, we investigated how Affinity and Diversity related to clean performance, but as pointed out by the reviewer did not comment on robust performance for which we made no claims. Though it was not the focus of the work, it is an interesting question!
>
> We have now taken the reviewers recommendation to expand the paper with additional experiments and a discussion of how affinity and diversity relate (or fail to relate) to robustness (see supplementary section G). In particular, we have evaluated robust performance on ImageNet-V2, CIFAR-10.1, and CIFAR-10-C. For the latter we present results for the overall CIFAR10-C robustness score, as well as accuracy for each of the composite corruptions. We think that this suggestion has strengthened the paper, but note that a truly thorough analysis of robustness is outside the scope of the current work.
>
> _In Figure 3, the highlights of the three augmentation methods, … are difficult to visualize._
>
> Sorry for the difficulty. We have replotted these augmentations in the updated manuscript. Please let us know if they are still hard to see.
>
> _In Figure 4, one diversity value (x-axis) seems to correspond to multiple test accuracy measurements._
>
> This is indeed correct! The plot illustrates that diversity alone is not a great predictor of accuracy, as the one-to-many relation shows. By looking at both affinity and diversity, this degeneracy is largely broken.
>
> _It seems that both positive and negative affinity values may help, according to Figure 3._
>
> We apologize if this was confusing. Affinity is always negative. Any appearance of positive affinity in Figure 3 is an artifact of the marker size.
>
> _high training loss may also mean a model still does not converge_
>
> Here it is important to note that we are comparing models trained with different augmentation strategies for equal training times. The models need not have converged to compare these relative loss values. As evidence to this point, in supplementary section G we show that the training loss early in training is also a good measure of Diversity.
>
> _It may be difficult for a new task with little experience to use the two metrics to select useful augmentation strategies._
>
> This is a valid point, and finding the most effective way to use the metrics to find successful strategies is ongoing work. We do note, however, that one or the other of these metrics are already individually being used successfully to search for novel augmentation strategies. For example, Fast AutoAugment and CTAugment focus on increasing the Affinity of an AutoAugment policy, whereas Adversarial AutoAugment focuses on increasing the diversity of an AutoAugment policy.
>
> Thank you again for your feedback! Are there any other concerns we didn’t address? If we have addressed your concerns, we humbly ask if you would kindly consider supporting accepting the paper by increasing the review score accordingly

---

### Comment · AnonReviewer4 · 2020-11-24
**About the definition of diversity**

I start this thread to potentially discuss with the authors about the definition of diversity (and affinity). This is independent from my review, it is not intended to produce any change in the manuscript, but to simply have a scientific discussion.

First, I would like to hear from the authors some more details about how they think their definition of diversity (final loss of a model trained with data augmentation) reflects the qualitative notion of diversity, as expressed in the paper, particularly in Figure 1c.

To me, the final loss of the model trained with data augmentation _is_ somehow related to diversity, but it seems that it does not capture all details of the concept. A measure of spread or entropy (discussed by the authors) seems closer, in my opinion.

One direction that has not been discussed would be defining affinity and diversity as a function of the model's activations (towards the transformed images of a data augmentation strategy). This entails some challenges, of course, but seems interesting to me. Have the authors explored this idea or thought about it?

---

> ### Author Response · Authors · 2020-11-24
> **re: About the definition of diversity**
>
> Thank you for the suggestion! Variance of activations does make intuitive sense, however a first pass at this doesn’t seem to correlate well with diversity, and instead correlates well with affinity (see Figure 21 in updated draft). A speculative reason for this is that the model trained only on clean data is not able to capture a meaningful notion of out of distribution diversity. But we agree making this work would be very interesting.
>
> As to the question of how training loss captures the notion of diversity depicted in figure 1c. In the cases where we had an alternative definition, such as entropy of the augmented data, we found empirically that models had more difficulty training on more complex (higher entropy) data -- represented by a higher training loss. This also matched our intuitive picture that more diverse data should be harder to fit. This intuition and the correlation with entropy lead us to use training loss to quantify diversity.
>
> Thanks again for taking the time to think about and discuss this, we sincerely appreciate the discussion, and would welcome any ideas now, or after the review period!

---

### Decision · Program_Chairs · 2021-01-07
**Final Decision**

**Decision:**

Accept (Poster)

**Comment:**

The reviews were largely split in the beginning. Although all reviewers find that the idea of diversity and affinity measures for data augmentation is intriguing and potentially useful, they also raised many concerns such as computationally expensive nature of the diversity metric, lack of clear methodology of how to utilize those metrics to design augmentation strategies in practice, and weak organization and presentation of some experiments which are seemingly less related to the main point of the paper. During the discussion phase authors made significant efforts to improve the paper, and some of the concerns are favorably addressed. As a result, two reviewers raised their initial scores, yet we still think the paper is on the borderline.

Overall, this paper presents an interesting and unique idea that potentially stimulate the community, while it also has some key weaknesses and has much room for improvements. Considering both pros and cons, we decided to accept the paper.